# Advances in Black-Box VI: Normalizing Flows, Importance Weighting, and Optimization

**Abhinav Agrawal[1], Daniel Sheldon[1,2], and Justin Domke[1]**

[1]College of Information and Computer Sciences, University of Massachusetts Amherst
[2]Department of Computer Science, Mount Holyoke College
{aagrawal, sheldon, domke}@cs.umass.edu

## Abstract

Recent research has seen several advances relevant to black-box variational inference (VI), but the current state of automatic posterior inference is unclear. One such advance is the use of normalizing flows to define flexible posterior densities for deep latent variable models. Another direction is the integration of Monte-Carlo methods to serve two purposes; first, to obtain tighter variational objectives for optimization, and second, to define enriched variational families through sampling. However, both flows and variational Monte-Carlo methods remain relatively unexplored for black-box VI. Moreover, on a pragmatic front, there are several optimization considerations like step-size scheme, parameter initialization, and choice of gradient estimators, for which there is no clear guidance in the literature. In this paper, we postulate that black-box VI is best addressed through a careful combination of numerous algorithmic components. We evaluate components relating to optimization, flows, and Monte-Carlo methods on a benchmark of 30 models from the Stan model library. The combination of these algorithmic components significantly advances the state-of-the-art "out of the box" variational inference.

## 1   Introduction

We consider the problem of automatic posterior inference. A scientist or expert creates a model $p(z, x)$ for latent variables $z$ and observed variables $x$. For example, $z$ might be population-level preferences for a political candidate in each district in a country, while $x$ is an observed set of polls. Then, we wish to approximate the posterior $p(z|x)$, i.e., determine what the observed data say about the latent variables under the specified model. The ultimate aim of automatic inference is that a domain expert can create a model and get answers without manually tinkering with inference details.

In practice, one often resorts to approximate inference. Markov chain Monte Carlo (MCMC) methods are widely applicable and are asymptotically exact, but sometimes slow. Variational inference (VI) approximates the posterior within a tractable family. This can be much faster but is not asymptotically exact. Recent developments led to "black-box VI" methods that, like MCMC, apply to a broad class of models [30, 15, 2].

However, to date, black-box VI is not widely adopted for posterior inference. Moreover, there have been several advances that are relevant to black-box VI, but have been little evaluated in that context. One such advance is normalizing flows, which define flexible densities through a composition of invertible transformations [31, 26]. Surprisingly, normalizing flows have seen almost no investigation for black-box VI; instead, they have been used either to directly learn a density [8, 9, 27, 19] or to bound the likelihood of a deep latent variable model $p_\theta(x) = \int p_\theta(z, x)dz$ [31, 20, 13]. In both cases, there is no mechanistic model and little or no focus on posterior queries.

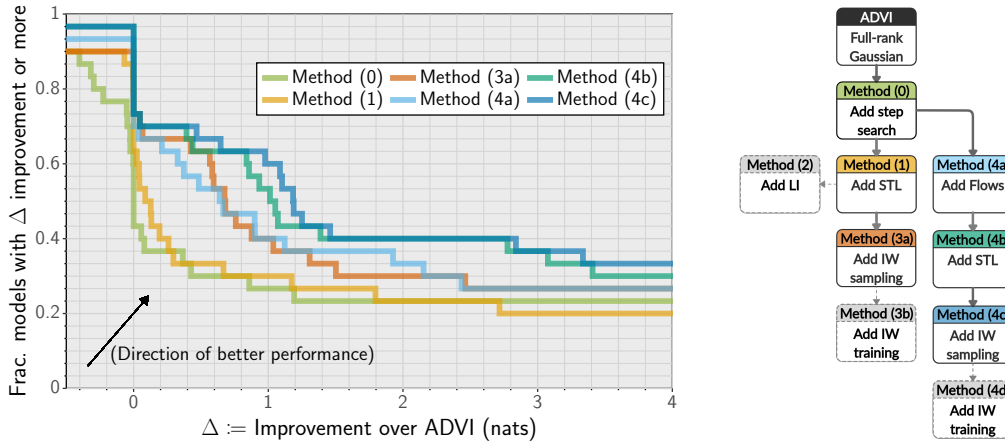

Figure 1: **(Left) Path study**: Empirical complementary-CDF for performance improvement over ADVI. **(Right) Relationship of methods**: Solid lines show modifications that improve performance (included on the left) dotted lines show modifications with unclear benefits (not shown on the left).

Another major direction is the development of tighter variational objectives that integrate Monte Carlo techniques such as importance weighting [3, 25, 23]. These were also initially designed to support learning of deep latent variable models. However, further developments have shown that they can also be viewed as enriching the variational family [1, 7, 10, 11]. Still, little is known about the practical performance of these techniques on real models. Further, importance weighting can be used in two ways. First, importance weighted *sampling* can be used at inference time to obtain better approximate samples, regardless of how the approximate distribution was created. Second, importance weighting *training* can be used during optimization to tighten an objective reflecting how well such a sampling scheme will work. There is little evidence in the literature about these.

It is essential for many users that black-box VI be *fully automatic*. This brings up numerous practical issues, such as initialization, step-size schedules, and the choice of gradient estimator. There have been few attempts to address this [21] and several basic questions remain unanswered: Do recent techniques to enrich variational families such as normalizing flows and variational Monte Carlo methods improve accuracy on mechanistic models? Which of these is more important? Can optimization be made robust enough to work with these techniques "out of the box" on real models?

The basic hypothesis of this paper is that automatic black-box VI can be best achieved through a thoughtful integration of many different algorithmic components. We carefully measure the impact of these components on 30 real models from the Stan model library [35, 36]. Our primary finding is that a combination of techniques to improve optimization robustness and enrich variational families collectively yield remarkable improvements in the accuracy of black-box VI. More concretely, we make the following observations.

- A prior step-size scheme often leads to suboptimal results. Using a comprehensive search over step-sizes is far more reliable (Figure 2).

- When using a Gaussian variational distribution, it is consistently helpful to use the sticking the landing" (STL) gradient estimator [33] that drops the "score term" from the evidence lower bound (ELBO) (Figure 3).

- Importance-weighted sampling can be used as a post hoc method to improve the posterior found by *any* inference method. This consistently improves results at minimal cost (Figures 4 and 8)

- Importance-weighted training introduces a trade-off between computational complexity and quality. The benefits are not consistent in our experiments (Figures 5 and 9).

- Real-NVP [9] (evaluated for the first time for black-box VI) gives excellent performance. Due to high nonlinearity, one might expect that fully-automatic optimization would be a challenge. However, when combined with our step-size scheme, it consistently delivers strong results without user-intervention (Figures 6 to 8).

- When using a normalizing flow variational distribution, the STL estimator is again helpful (Figure 7). However, this uses an inverse transform, which is not efficient with all flows (Section 4.3).

- The doubly-reparameterized estimator [37] consistently improves convergence when applied to importance weighted training (Figure 13 in appendix). However, as above, the overall value of importance-weighted training is unclear.

Our final method combines importance sampling, normalizing flows, a highly robust step-size scheme and the STL gradient estimator, to improve the performance by one nat or more on at least 60% of the models when compared to Automatic Differentiation Variational Inference (ADVI) (see performance of Method (4c) in Figure 1). We believe this strategy represents the state-of-the-art for fully-automatic black-box VI for posterior inference.

## 2 Problem Setup

Directly computing the posterior is typically intractable; VI searches for the closest approximation of $p(z|x)$ within a parameterized family $q_\phi(z)$ by maximizing the ELBO [34]

$$\mathcal{L}(\phi) = \mathbb{E}_{q_\phi(z)}\left[\log p(z, x) - \log q_\phi(z)\right] = \log p(x) - \mathbb{KL}[q_\phi(z)\|p(z|x)]. \tag{1}$$

Since KL-divergence is non-negative, $\mathcal{L}(\phi)$ lower-bounds $\log p(x)$ for all $\phi$. Moreover, when $p$ is fixed, optimizing $\mathcal{L}(\phi)$ is equivalent to minimizing the KL-divergence between $q_\phi(z)$ and $p(z|x)$.

Exploiting the observations above, there are two basic uses of VI. The first is to *learn* the parameters $\theta$ of a model $p_\theta(z, x)$ from examples of $x$. In this case, VI is used to provide a tractable objective that lower-bounds the exact likelihood. This is the basis of variational auto-encoders [18, 32] and their relatives. The second use of VI is to *infer* the posterior $p(z|x)$ of a fixed model. This is the setting of black-box VI [30], the focus of this paper.

### 2.1 Rules of engagement

This paper studies many black-box VI methods. Each method is optimized using its own objective to produce an approximate posterior $q_\phi(z)$. During optimization, all methods have the same computational budget, measured as 100 "oracle evaluations" of the $\log p$ per iteration, and are optimized for 30,000 iterations. Then, 10,000 fresh samples are drawn from $q_\phi$ to evaluate either the ELBO in Equation (1) or the importance-weighted ELBO in Equation (4), depending on the method. In either case, this gives a lower-bound on $\log p(x)$ (or equivalently an upper-bound on the KL-divergence [10]). Applying importance-weighting can be seen as an *algorithmic component*, not just a "metric" since it gives a different approximation of the posterior [11].

We evaluate each method using a benchmark of 30 models from the Stan Model library [35, 36]. To remove any ambiguity, a standalone description of each method compared is given in Appendix A.

Visualizing results across many models and inference schemes is a challenge. Simple ideas like scatterplots fail because there are many inference schemes and huge differences in $\log p(x)$ across models. Instead, we study results using "empirical complementary CDF plots". Two inference methods are compared by computing, for each model $i \in \{1, \cdots, 30\}$, the difference $\Delta_i$ between the lower-bounds produced by the two methods. The idea is to plot, for each value of $\Delta$, the fraction of values $\Delta_i$ that are $\Delta$ or higher. This shows how often a given inference method improves on another by a given magnitude $\Delta$.

We start from a simple ADVI baseline model and consider changes in a single algorithmic component one by one in Figures 2 to 9. Next, we compare the full "path" of beneficial components to ADVI in Figure 1. Finally, we perform an ablation study removing individual algorithmic components from the final best system in Figure 10. We conduct three independent trials for all of our experiments. For space, the path and ablation study show only a single trial in the main text, with others in the supplement. We also conduct additional experiments in Appendix B including comparisons between diagonal and full-rank Gaussian VI and more comparisons of different gradient estimators.

# 3 Optimization in BBVI

Most uses of black-box VI make optimization choices on a model-specific basis. A notable exception to this is the ADVI [21], which we use as our baseline. ADVI is integrated into Stan, a state-of-the-art probabilistic programming framework for defining statistical models with latent variables [4]. This makes ADVI one of the most widely available black-box VI algorithms. There are four key ideas: (1) to automatically transform the support of constrained random variables to an unconstrained space (2) to approximate the resulting unconstrained posterior with a Gaussian variational distribution, (3) to estimate the gradient of the ELBO using the reparameterization trick and (4) to do stochastic optimization with a fully-automated step-size procedure.

In all cases, we use the idea of transformation to unconstrained domain unchanged (1). In the rest of this section, we consider modifications to optimization ideas (3,4). In the next section, we consider generalization beyond Gaussians (2).

## 3.1 ADVI step size search

ADVI uses a novel decreasing step-size scheme based on a base step-size $\eta$ (we review this in Appendix D). In short, for each $\eta \in \{0.01, 0.1, 1, 10, 100\}$, optimization is run for a small number of iterations, and the value that provides the highest final ELBO value when estimated on a fresh batch of samples is used. We use a batch of fresh 500 samples after 200 iterations. It is not clear to what degree results after a small number of iterations can give a realistic picture of how optimization will look after a large number of iterations. To test this, we propose a straightforward alternative that simply runs different steps more exhaustively.

## 3.2 Comprehensive step search

We perform optimization using Adam [17], and we comprehensively search for the Adam step-size $\eta$ in the range $\frac{0.1}{D}[1, B^{-1}, B^{-2}, B^{-3}, B^{-4}]$, where $D$ is the number of latent dimensions of the model and $B$ is a decay constant; we use $B = 4$. For each step size in this range, we optimize for a fixed number of iterations while keeping the step size constant. The number of iterations is constrained only by the sampling/computational budget of a user; we use 30,000 iterations for each step-size choice. Finally, we select the parameters from the step size that led to the best average-objective, averaged over the entire optimization trace. We found this to be a more robust indicator of performance than estimating the final ELBO using a smaller batch of fresh samples (see Appendix F for discussion).

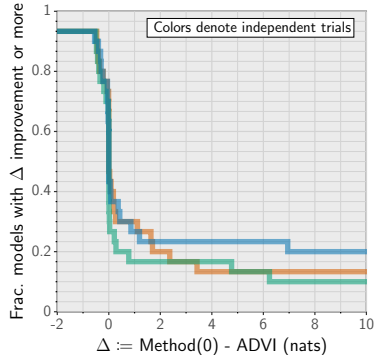

Figure 2: Using comprehensive step-size search provides significant gain of 1 nat on almost 20% of the models (high variability is due to ADVI).

Figure 2 shows that the comprehensive search provides a significant improvement of 10 nats or more on a small fraction of the models (both schemes in the figure use the closed-form entropy estimator from Equation (2)).

## 3.3 Gradient Estimation

Most black-box VI methods are based on the reparameterization trick. Suppose that $q_\epsilon$ is a fixed distribution and $z_\phi(\epsilon)$ a function such that if $\epsilon \sim q_\epsilon$ then $z_\phi(\epsilon) \sim q_\phi$. Then, if $H(\phi)$ is the entropy of $q_\phi$, the gradient of Equation (1) can be written as

$$\nabla_\phi \mathcal{L}(\phi) = \mathop{\mathbb{E}}_{q_\epsilon(\epsilon)} \left[ \nabla_\phi \log p(z_\phi(\epsilon), x) \right] + \nabla_\phi H(\phi). \tag{2}$$

The gradient can be estimated by drawing a single $\epsilon$ (or a minibatch). In some cases (e.g., Gaussians) $H$ is computed in closed form, so the above equation can be used unchanged; this is estimator used in ADVI. Alternatively, $\nabla H$ can also be estimated using the reparameterization trick, using either of

$$\nabla_\phi H(\phi) = \underset{q_\epsilon(\epsilon)}{\mathbb{E}} \underbrace{\nabla_\phi \log q_\phi(z_\phi(\epsilon))}_{\text{"Full" estimator}} = \underset{q_\epsilon(\epsilon)}{\mathbb{E}} \underbrace{(\nabla_\phi \log q_\theta(z_\phi(\epsilon)))_{\theta=\phi}}_{\text{"STL" estimator}}. \quad (3)$$

The second option, known as STL estimator [33], "holds $\phi$ constant" under the gradient. This is valid since $\mathbb{E}_{q_\phi(z)} \nabla_\phi \log q_\phi(z) = 0$.

Figure 3 compares the results of the STL estimator to a closed-form entropy. The STL estimator is usually preferred. We give a more comprehensive comparison of estimators in Figure 12 (supplement).

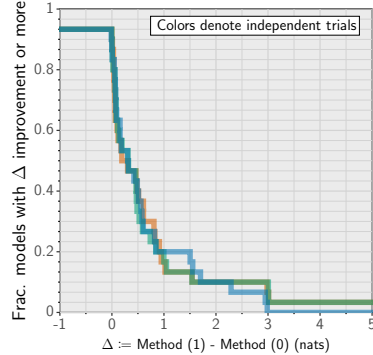

### 3.4 Intialization

Initialization can have a major influence on convergence. ADVI initializes to a standard Gaussian. A natural alternative for Gaussian or Elliptical variational distributions is to use Laplace's method to initialize parameters [10, 11]. This method uses black-box optimization to find $\hat{z}$ to maximize $\log p(z, x)$, computes the hessian of $\log p$ at $\hat{z}$ and then sets $q_\phi(z)$ to be the Gaussian whose log-density matches the local curvature of $\log p(\hat{z}, x)$. While Laplace's method is intuitive, it could conceivably be harmful, e.g. by providing an initialization that leads to a worse local optima.

Figure 3: Adding STL to Gaussian VI is consistently helpful across the models and hurts only in minority of cases.

To the best of our knowledge, there are no existing studies of these initialization schemes. Figure 14 (supplement) uses the previous comprehensive step search and compares the results of adding Laplace's initialization (LI). A similar fraction of models are helped and harmed by the change.

## 4 Enriched Variational Families

### 4.1 Monte Carlo Objectives

VI tries to approximate $p(z|x)$ with $q_\phi(z)$. If the approximation is inexact, Monte Carlo methods can often improve it. Burda et al. [3] introduced the "importance-weighted" ELBO (IW-ELBO) to use in place of the conventional ELBO when training a latent-variable model. This is

$$\mathcal{L}_M(\phi) = \underset{q_\phi(z_1)\cdots q_\phi(z_M)}{\mathbb{E}} \left[ \log \frac{1}{M} \sum_{m=1}^M \frac{p(x, z_m)}{q_\phi(z_m)} \right] \le \log p(x). \quad (4)$$

Here, $z_1 \dots z_M$ are iid samples from $q_\phi(z)$. This reduces to the standard ELBO when $M = 1$. The IW-ELBO increases with $M$ and approaches $\log p(x)$ asymptotically.

The IW-ELBO is a measure of the accuracy of self-normalized importance sampling on $p$ using $q_\phi$ as a proposal distribution. Define $q_{M,\phi}$ to be the distribution that results from drawing $M$ samples from $q_\phi$ and then selecting one in proportion to the self-normalized importance weights $p(z_m, x)/q(z_m)$. Then, $\mathcal{L}_M$ is a relaxation of the ELBO defined between $q_{M,\phi}$ and the target $p$ [1, 7, 25]. Alternatively, $\mathcal{L}_M$ can be seen as a traditional ELBO between distributions that *augment* $q_{M,\phi}$ and $p$ [10].

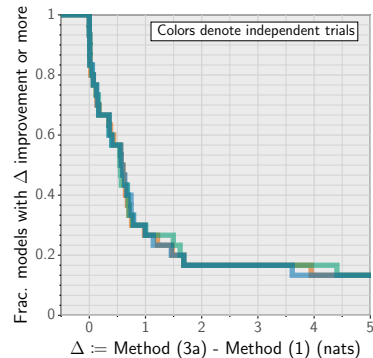

For posterior approximation, importance weighting can be applied in two ways.

Figure 4: IW-sampling greatly improves the results of Gaussian VI.

1. **Importance-weighted sampling**. For *any* distribution, importance-weighting can be applied at test time to improve the quality of the posterior approximation. We know this because $\mathcal{L}_M(\phi) \ge \mathcal{L}(\phi)$ and thus $\mathbb{KL}[q_{M,\phi}(z)\|p(z|x)] \le \mathbb{KL}[q_\phi(z)\|p(z|x)]$ [10]. This should not be seen as a metric for evaluating $\phi$. Rather, it is an algorithmic component of inference, since it yields *different samples* that are actually closer to the posterior [11].

2. **Importance-weighted training**. To find the parameters $\phi$, one can optimize the IW-ELBO. The idea is that if one intends to perform importance-weighted sampling, this directly optimizes for parameters $\phi$ that will perform well at this task. That is, optimizing the IW-ELBO implicitly makes $q_{M,\phi}(z)$ close to $p(z|x)$.

Figure 4 shows the results of adding importance-weighted sampling to the previous model, with $M = 10$. This produces a significant benefit of 1 nat or more on 30% of models and never hurts. This also comes at a minimal computational cost since there is no modification of the training procedure. Considerations in importance-weighted training are a bit subtle and discussed next.

### 4.2 Importance-weighted training

Optimizing Equation (4) requires a gradient estimator. The most obvious estimator is

$$\nabla \mathcal{L}_M(\phi) = \mathop{\mathbb{E}}_{q_\epsilon(\epsilon_1)\cdots q_\epsilon(\epsilon_M)} \sum_{m=1}^{M} \hat{w}_m \nabla_\phi \log \frac{p(z_\phi(\epsilon_M), x))}{q_\phi(z_\phi(\epsilon_m))}, \quad (5)$$

where $\hat{w}_m = \frac{w_m}{\sum_{l=1}^{M} w_l}$ for $w_m = \frac{p(x, z_\phi(\epsilon_m))}{q_\phi(z_\phi(\epsilon_m))}$. However, Rainforth et al. [28] point out that the signal-to-noise-ratio of this gradient estimator scales as $1/\sqrt{M}$, suggesting optimization will struggle when $M$ is large. To circumvent this problem, Tucker et al. [37] introduced the "doubly reparameterized gradient estimator" (DReG), which does not suffer from the same issue. This is based on the representation of

$$\nabla \mathcal{L}_M(\phi) = \mathop{\mathbb{E}}_{q_\epsilon(\epsilon_1)\cdots q_\epsilon(\epsilon_M)} \sum_{m=1}^{M} (\hat{w}_m)^2 \nabla_\phi \log \frac{p(z_\phi(\epsilon_m), x))}{q_\theta(z_\phi(\epsilon_m))} \Bigg|_{\theta=\phi}. \quad (6)$$

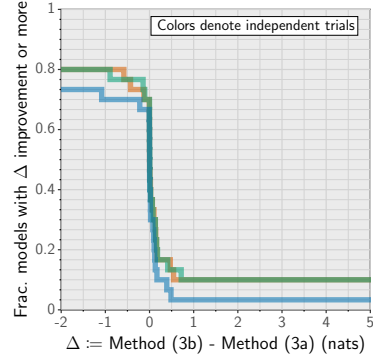

Figure 5: Adding IW-training to IW-sampling offers no clear advantage with Gaussians.

Intuitively, the idea is that since $\theta$ is $\phi$ "held constant" under differentiation, the variance is reduced. The difference of the above two estimators is analogous to the difference of the two estimators of the entropy gradient in Equation (3).

We found the DReG out-performed the estimator from Equation (5); using it added 1 nat of improvement to almost 30% of the models (see Figure 13, supplement). However, the benefits are not consistent when compared to IW-sampling alone. Figure 5 shows the performance *decays* on around 20% of models. Thus, when keeping evaluations of $\log p$ fixed at our budget, better performance results from training the regular ELBO with the STL estimator and then using IW-sampling only.

### 4.3 Normalizing Flows

Normalizing flows construct a flexible class of densities by applying a sequence of invertible transformations to a simple base density. Rezende and Mohamed [31] first introduced normalizing flows in the specific context of approximating distributions for VI; they have since been studied in a much more general context (e.g., for density estimation [9, 19, 13]).

The main idea is to transform a base density $q_\epsilon(\epsilon)$ using a diffeomorphism $T_\phi$. The variable $z = T_\phi(\epsilon)$ has the distribution

$$q_\phi(z) = q_\epsilon(\epsilon)|\det \nabla T_\phi(\epsilon)|^{-1} \quad (7)$$

$$= q_\epsilon(T_\phi^{-1}(z))|\det \nabla T_\phi^{-1}(z)|. \quad (8)$$

Typically, $T$ is chosen as a sequence of transforms, each of which is parameterized by a neural network. These transforms are designed so that the determinant of the Jacobian $|\det \nabla T_\phi(\epsilon)|$ can be computed efficiently.

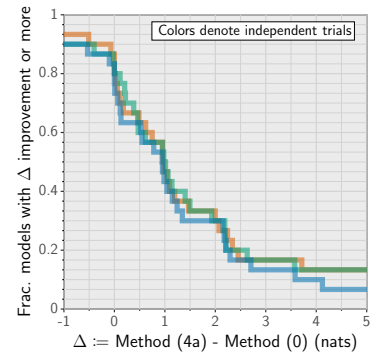

Figure 6: Substituting flows for Gaussians improves half of the models by 1 nat or more.

We use real-NVP–a coupling-based flow [9] with 10 successive transformations, each determined by a fully-connected network with 2 hidden layers each with 32 hidden units (see Appendix E for full details). Figure 6 shows the results of using normalizing flows instead of Gaussians on the benchmark. Since there is no closed-form entropy, these results estimate it using the middle estimator from Equation (3). This yields a significant improvement of 1 nat or more on around half of models.

Depending on the gradient estimator used, the inverse of $T_\phi$ may or may not be needed. This issue is somewhat subtle. In general, to evaluate the density $q_\phi$ at an arbitrary point requires computing the inverse $T_\phi^{-1}$ which may be inefficient [16, 6]. While the ELBO in Equation (1) requires evaluating the density $\log q_\phi$ it is only done on a point sampled from $q_\phi$. This means this can be done using only $T_\phi$–by first sampling $\epsilon$ and then transforming to $z$ there is no need to explicitly compute an inverse. Similarly, if one is to estimate the gradient of the entropy using the "full" estimator (middle equation in Equation (3)), only $T_\phi$ is needed. However, the estimator that drops the score term (right equation in Equation (3)) is typically lower variance. The natural way to do this is to first sample from $q_\phi$, and then, in a second *independent* step, evaluate the sample at parameters $\theta = \phi$ that are held constant under differentiation. In this second step, the sample is treated as an arbitrary point so that derivatives will only flow to the sampling part of the algorithm. This requires the inverse $T_\phi^{-1}$.

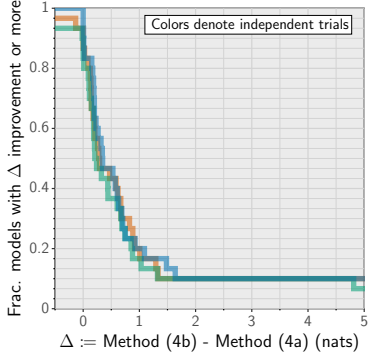

Figure 7: Using the STL estimator for flows (instead of the "full" estimator) is typically helpful and gives huge improvements on a minority of models.

With real-NVP flows, both the forward and reverse transform are efficient, so the second estimator from Equation (3) can be used. Figure 7 compares the results of substituting this estimator instead. In many cases, the results are much the same. However, in a significant minority of models, the new estimator yields enormous improvements.

## 4.4 Importance Weighting with Normalizing Flows

Importance weighting and normalizing flows can be applied together. The ideas in Section 4.1 are not specific to Gaussian densities. In the first experiment, we take the same optimization scheme used above, and apply importance weighted sampling only (with no change to the optimization procedure). As shown in Figure 8, this change never hurts, provides small improvements for most models, and provides large improvements on a minority of models. Since this again comes at a minimal cost (extra work is only needed in the sampling stage) it is very worthwhile.

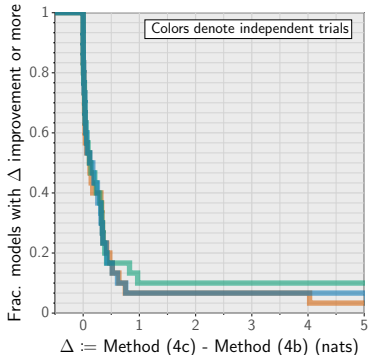

Figure 8: Adding IW-sampling to normalizing flows can significantly improve some of the models and never hurts.

One can also explicitly optimize the IW-ELBO. Figure 9 demonstrates that adding IW-training provides large benefits on a few models but hurts performance by 1 nat or more for almost 10% of the cases. These observations mirror the effect observed for Gaussians in Figure 5.

We now arrive at our best method: train normalizing flows with a regular ELBO objective using the lower-variance STL gradient and then add importance sampling only during inference. This is easily the go-to black-box VI strategy when we fix a budget for $\log p$ evaluations.

## 5 Experiments

**Using Stan with auto-diff packages:** Our experiments are based on a set of models (described in Table 1 in supplement) from the Stan Model library [36, 35]. In order to test our VI variants, we designed a simple interface to use Stan models with VI algorithms implemented in Autograd, a Python automatic differentiation library [22] (refer to Appendix C for more details)

**Architectures:** For normalizing flow methods, we use real-NVP flow with 10 coupling layers with a fixed base architecture for all models. Complete details are present in Appendix E.

For experiments that use a full-rank Gaussian variational family, we parameterize the mean $\mu$ and the Cholesky factor $L$ of the covariance matrix $\Sigma$, such that $\Sigma = LL^\top$. Parameters are initialized to a standard normal when LI is not used.

**Laplace's Initialization:** We use SciPy's BFGS optimize routine to maximize $\log p(z, x)$ over $z$ to get $\hat{z}$; we optimize for 2000 iterations (with default settings for other hyper-parameters) [38]. At $\hat{z}$, we compute the Hessian matrix $H$ using two-point finite differences with $\epsilon = 10^{-6}$. To use LI, we set $\mu = \hat{z}$, and set $L$ such that $LL^\top = (-H)^{-1}$.

**Training procedure:** To maintain a fair "oracle complexity" in terms of evaluations of $\log p$, each gradient estimate averages over a batch of independent gradient estimates, such that there are always 100 total evaluations of $\log p$ in each iteration. To use Figure 9 as an example, IW-training method averages 10 different copies of the DReG estimator at each iteration (M = 10), and IW-sampling method averages 100 different copies of STL estimator (can be viewed as M = 1). For all experiments, we use the comprehensive step-size search scheme mentioned in Section 3. Wherever importance weighting is used (sampling or training), $M = 10$

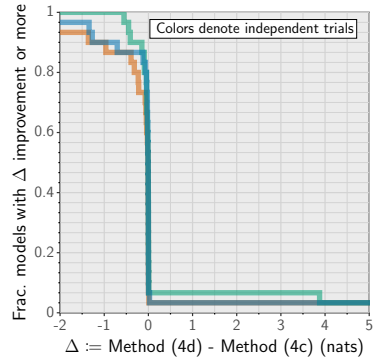

Figure 9: Adding IW-training to IW-sampling offers little advantage with flows.

**Replicating ADVI:** For a fair comparison, we re-implement ADVI optimization in our own framework (refer to Appendix D for more details). In our preliminary experiments found that the performance matched the PyStan version for the same hyper-parameter settings.

**Metric:** To compare the performances, we use a fresh batch of 10,000 samples. Again, to maintain a fair "oracle complexity", we use 10,000 samples irrespective of the $M$ used for importance weighting. To use Figure 10 as an example, when we ablate IW-sampling, we average 10,000 copies of ELBO estimate whereas the "Best model" uses the 1,000 copies of the IW-ELBO estimate (M=10).

**Results:** We provide the complete tables of results with final-metric values from the independent trials in Appendix H in the supplement.

## 5.1 Path Study

We summarize the useful algorithmic changes in a path-study by comparing against the common ADVI baseline. Figure 1 presents results from independent trial (see Figure 15 in supplement for full results). Across the trials, we see that with each step in the path performance improves, and the final strategy provides significant improvement of 1 nat or more on at least 50% of the models.

## 5.2 Ablation Study

We analyze the performance of the "best" variational approach by taking away each of it's components individually. Figure 10 presents the ablation study for an independent trial (see Figure 16 in supplement for full results). Across the trials, each component is complementary–they add to the other and the combination performs the best.

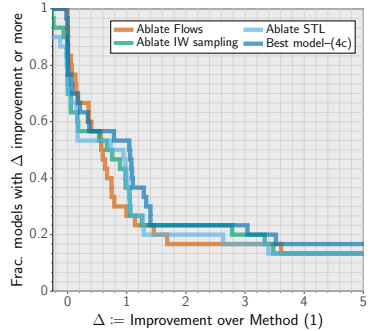

Figure 10: **Ablation Study:** Removing flows has a strong impact; removing STL or IW-sampling has a lesser impact on the performance.

# 6  Discussion

**Related work.**    Fjelde et al. [14] develop a software package (Bijector.jl) for bijective variable transformation that works independent of the computational framework. As a demonstration, they use normalizing flows to relax the mean-field assumption; they add coupling layers on top of a learnable diagonal Gaussian. However, the aim of our work is different–we examine the efficacy of recent inference ideas for automating black-box VI. Parallel to our work, Webb et al. [39] improved automatic inference by extending ADVI with normalizing flows.  In comparison, our analysis goes beyond flows and studies the interaction of various algorithmic components, and our careful experimental framework provides several valuable insights (see observations in Section 1.)

**General comments.**    We choose ELBO/IW-ELBO to compare different VI methods. Several works have reported an empirical correlation between the ELBO improvements with improvements in test-likelihoods (see [23, Appendix, Table 3, 4, and 5]; [5, Figure 4]; [24, Figure 4].) and accuracy of posterior moments (see [10, Figure 6]; [11, Figure 7, and Figure 9 to 22]).) We expect our methods to follow similar correlations for ELBO improvements.

In our initial experiments, the performance correlated well with the number of flow layers and the number of hidden units; however, we do not aim to find the best possible flow. We use a reasonably generic real-NVP flow that allows for efficient forward and inverse; efficient inverse affords for the use of STL/DReG gradient. The use of better flow variants and optimal architectures can hopefully improve on our work.

**Computations costs.**    Trade-offs exist between the improvement in performance and the computational costs we are prepared to entertain. With more computing resources, better performance is possible.

A comprehensive step-search scheme can be done in parallel. Adding Laplace Initialization requires one to solve a maximization problem beforehand. Adding Importance Weighting training, with a fixed "Oracle budget" for querying $\log p$, may seem to be relatively inexpensive; however, the posterior inference scales linearly in $M$.

Normalizing flows require more memory and computation. The parameters for full rank Gaussian scale quadratically in the number of dimensions and scale linearly for coupling-based models we consider; flows involve large constants and are expectedly slower for lower-dimensional models. Further, using the DReG or STL calculation comes at an additional cost for normalizing flows due to a second pass of $T_\phi^{-1}$ (see Section 4.3 for details.)

**Conclusions.**    This paper provides clear empirical evidence that systematically combining recent advances can achieve a robust inference scheme.  Using step-search instead of ADVI's heuristic step selection and using normalizing flows instead of full-rank Gaussian is immensely helpful; the STL/DReG gradient consistently outperforms other available options; using Importance Weighted sampling is an almost always beneficial post-hoc step. Under the fixed sample budget, we did not find Importance Weighted training and Laplace's initialization to be consistently helpful. We hope that lessons from our work can help both researchers and practitioners.

# Broader Impact

Automatically inferring hidden variables in these models is a challenge to date. Our approach and observations can prove instrumental for researchers in various fields–epidemiologists, ecologists, political scientists, social scientists, experimental psychologists, and others.  While some other inference methods may perform better for a particular model, our ideas provide a formidable baseline.

Our framework combines several ideas–for a practitioner, this can make diagnosing the source of sub-optimality tricky. While it is possible to conduct comprehensive searches for hyper-parameters, the ability to do so hinges on access to adequate computation resources.

## Acknowledgments and Disclosure of Funding

This material is based upon work supported by the National Science Foundation under Grant No. 1908577.

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
