[Supplementary Material]

# A Full Method Description

**ADVI Baseline**

Uses a full-rank Gaussian initialized to standard normal and optimizes with closed-form entropy gradient from Equation (2); uses ADVI step-scheme for updates (see Appendix D for more details). Importance-weighted sampling is not used; importance-weighted training is not used (optimized standard ELBO).

**Method (0)**

Uses a full-rank Gaussian initialized to standard normal and optimizes with closed-form entropy gradient from Equation (2); uses our comprehensive step-size search for updates (see Section 3.2 for more details). Importance-weighted sampling is not used; importance-weighted training is not used.

**Method (1)**

Uses a full-rank Gaussian initialized to standard normal and optimizes with the STL from Equation (3); uses our comprehensive step-size search for updates. Importance-weighted sampling is not used; importance-weighted training is not used.

**Method (2)**

Uses a full-rank Gaussian and initializes with LI method from Section 3.4; optimizes with the STL from Equation (3) and uses our comprehensive step-size search for updates. Importance-weighted sampling is not used; importance-weighted training is not used.

**Method (3a)**

Uses a full-rank Gaussian initialized to standard normal and optimizes with the STL gradient from Equation (3); uses our comprehensive step-size search for updates. Importance-weighted sampling is used with $M = 10$; importance-weighted training is not used.

**Method (3b)**

Uses a full-rank Gaussian initialized to standard normal and uses importance-weighted training with M = 10; optimizes with the DReG from Equation (6) and uses our comprehensive step-size search for updates. Importance-weighted training is used with M = 10 (optimizes IW-ELBO with M = 10).

**Method (4a)**

Uses a real-NVP normalizing flow (see Appendix E for architectural and initialization details) as $q_\phi$; optimizes with the "full" gradient from Equation (3) and uses our comprehensive step-size search for updates. Importance-weighted sampling is not used; importance-weighted training is not used.

**Method (4b)**

Uses a real-NVP normalizing flow and optimizes with the STL gradient from Equation (3); uses our comprehensive step-size search for updates. Importance-weighted sampling is not used; importance-weighted training is not used.

**Method (4c)**

Uses a real-NVP normalizing flow and optimizes with the STL gradient from Equation (3); uses our comprehensive step-size search for updates. Importance-weighted sampling is used with M = 10; importance-weighted training is not used.

**Method (4d)**

Uses a real-NVP normalizing flow and uses importance-weighted training with M = 10; optimizes with the DReG gradient from Equation (6) and uses our comprehensive step-size search for updates. Importance-weighted training is used with M = 10.

# B  Extended results

## B.1  Diagonal vs Full-rank Gaussian VI

In this section, we compare the performance of Gaussian VI with full-rank covariance against diagonal covariance. While it is well known that full-rank covariance Gaussian distribution are more expressive, a clear experimental evidence for this is notably missing from the literature–we supplement this by experimenting with three methods from our path-study: Method (0), Method (1), and Method (3a). In Figure 11, it is easy to observe that using full-rank Gaussian improves performance by 1 nats or more on at least half of the models across the methods. When using Importance Weighted sampling–Method (3a)– full-rank covariance Gaussians almost always improves the performance.

Figure 11: Full-rank vs. Diagonal: (a) Method (0): closed-form entropy w/ step search, (b) Method (1): we replace closed-form entropy with STL gradient, and (c) Method (3a): we add Importance Weighted Sampling to STL gradient. In all the methods, using full-rank Gaussian improves the performance by at least 1 nats on more than half of the models.

## B.2  Different gradient for Gaussian VI

There are three choices of gradients for the Gaussian family. First, as Gaussians have a closed-form entropy, we can use the gradient from Equation (2); this is the gradient that ADVI uses. Second, we can alternatively use the middle gradient from Equation (3). Third, we can drop the score-function term and use the STL estimator (third term in Equation (3)). In Figure 12, the first panel compares the performance of using STL against the ADVI implementation (ADVI step-scheme and gradient). In the second panel, we compare the performance with the closed-form estimator optimized using our comprehensive step-search. Finally, we compare against the middle gradient in Equation (3). In all the alternatives, STL rarely hurts and adds significant value to several models.

## B.3  IW-training with DReG

We compare IW-training with and without DReG estimator on different possibilities and find that it consistently improved the performance. In Figure 13, we first compare the performance with the standard IW-ELBO gradient for Gaussian families. In the second comparison, we add Laplace Initialization to both methods, IW-ELBO gradient and DReG gradient. In the final comparison, we

Figure 12: (a) STL against ADVI; STL improves the performance by 1 nat or more on almost 30% of the models. (b) Next, we replace the ADVI step-size scheme with our comprehensive step search. STL improves the performance on 20% of the models by 1 nat or more. (c) We also compare against the middle gradient from Equation (3) and find that STL provides an improvement of 1 nat or more on almost 10% of the models. In all the alternatives, STL rarely hurts.

compare DReG with regular IW-ELBO gradient for normalizing flows. Across all comparisons, DReG improves the performance and rarely hurts.

Figure 13: (a) DReG improves the performance significantly by 1 nat on almost 30% of the models for Gaussians when initialized with standard normal (b) On adding LI to the previous model, DReG adds 1 nat to around 20% of the models (c) Adding DReG to IW-training of flows also helps. We observe significant improvement of 1 nat or more for almost 30% of the models

## B.4 Path Study - full results

We conduct a path-study to accumulate all the useful combinations of our analysis. Figure 15 presents the study for three independent trials. The high variation is due to the ADVI; on 10 models out of 30, ADVI diverges in at-least one trial for our implementation. If an optimization diverges, we set the improvement as zero, that is, we count the model in favor of the baseline (see Table 2 for values).

## B.5 Ablation Study - full results

We conduct an ablation-study to analyze each component of the best performing method. Figure 16 presents the study for three independent trials.

Figure 14: Adding LI to Gaussian VI is neither consistently helpful nor consistently harmful.

Figure 15: Across the trials: Method (1) that uses STL gradient improves over ADVI by 1 nat or more for at least 20% of the models. Method (3a) adds the IW-sampling to (1) and improves by a nat or more on at least 30% of the models. Method (4a) uses flow with the naive gradient estimator and achieves performance similar to (3a). Method (4b) adds the STL gradient to (4a) and improves on at least 40% of the models by 1 nat or more. Method (4c) adds IW-sampling to (4b) and improves by 1 nat on a minimum of 50% of the models. All our methods use comprehensive step-search and use M = 10 wherever IW-sampling is applied.

## B.6 Laplace Initialization

Figure 14 compares the results of using Laplace initialization (LI) against not using it (we omitted this comparison from the main text for brevity). While there is a significant improvement on a minority of models, similar fraction observe a significant decay.

Figure 16: Across the trials: ablating STL observes the least decay in performance while ablating flows causes the most decrease. The effect of ablating IW-sampling lies somewhere in the middle of these two. All approaches are trained with comprehensive step-search and use M=10 wherever importance weighted sampling is used.

## C   Interfacing using auto-diff packages

To interface with Stan models, we must define a new "primitive" function in Autograd that corresponds to $\log p(x, z)$ as a function of $z$. In addtion, this also requires computing $\log p(x, z)$ itself as well as the gradient-vector product $a^\top \nabla_z \log p(z, x)$ for any vector $a$. This is easily done since PyStan interface allows access to $\log p(z, x)$ and the gradient $\nabla_z \log p(z, x)$ for any model defined in Stan. This approach has the disadvantage that high-order gradients are not possible. Similar strategies could be used with other automatic differentiation packages.

# D ADVI

**Step-size scheme**  ADVI uses a novel step-size sequence inspired by adaptive step-size gradient schemes [12, 29, 17]. The update at iteration $i$ is

$$\phi^{(i+1)} = \phi^{(i)} - \rho^{(i)} \odot g^{(i)}, \tag{9}$$

where $g^{(i)}$ is the stochastic gradient in the $i$-th iteration, $\rho^{(i)}$ is a vector of step-sizes (one per coordinate of $\phi$) and $\odot$ denotes elementwise multiplication. To determine the stepsizes, a vector $s^{(i)}$ is initialized to $s^{(1)} = (g^{(1)})^2$ and maintained recursively as

$$s^{(i)} = \alpha(g^{(i)})^2 + (1 - \alpha)s^{(i-1)}, . \tag{10}$$

Then, the stepsizes are chosen as

$$\rho^{(i)} = \frac{\eta}{i^{1/2+\epsilon} \times \left(\tau + \sqrt{s^{(i)}}\right)}, \tag{11}$$

where the square root and division are element-wise. Here $\eta > 0$ is the scale of the step-size, $i^{1/2+\epsilon}$ decays the step over time, and the $s^{(i)}$ adapts the curvature of the ELBO. $\tau = 1$ and $\epsilon = 10^{-16}$ are stabilizing constants.

**Implementation details**  ADVI step-scheme search for $\eta$ from Equation (11) over the range $\{0.01, 0.1, 1, 10, 100\}$ to best adapt to the size of the problem. We use 200 optimization iterations for each of these choices and then use a fresh batch of 500 samples for each step in the range to calculate final ELBO values. The step with highest final ELBO is selected as the adapted step-size; with the adapted $\eta$ we optimize for 30,0000 iterations where, at each iteration we use 100 total $\log p$ evaluation(same as our other experiments).

We also implement the relative-tolerance convergence criterion implemented in PyStan to detect early convergence(we use a tolerance of 0.001). Also, following the original work, we use the closed form of entropy of $q_\phi$ for the ADVI training objective. We make an honest attempt to the best of our abilities to re-implement the ADVI and in our preliminary experiments found that the performance matched the PyStan version for the same hyper-parameter settings. We found that the performance of ADVI was highly variable; out of the three independent trials, 9 models diverged in at least one trial. Replacing ADVI step-scheme with our comprehensive step-search saw no divergence for the closed-form entropy case that uses Adam optimizer.

# E  Implementation details for real-NVP

**Architectural details:**  We use a real-NVP flow with 10 coupling layers for all our experiments. We define each coupling layer to be comprised of two transitions, where a single transition corresponds to affine transformation of one part of the latent variables. For example, if the input variable for the $k^{th}$ layer is $z^{(k)}$, then first transition is defined as

$$\begin{aligned} z_{1:d} &= z_{1:d}^{(k)} \\ z_{d+1:D} &= z_{d+1:D}^{(k)} \odot \exp\left(s_k^a(z_{1:d}^{(k)})\right) + t_k^a(z_{1:d}^{(k)})). \end{aligned} \tag{12}$$

where, super-script $a$ denotes first transition and sub-script $k$ denotes the $k^{th}$ layer. For the next transition, the $z_{d+1:D}$ part is kept unchanged and $z_{1:d}$ is affine transformed in a similar fashion to obtain the layer output $z^{(k+1)}$(this time using $s_k^b(z_{d+1:D}^{(k)})$ and $t_k^b(z_{d+1:D}^{(k)})$ ). This is also referred to as the alternating first half binary mask. Both, scale($s$) and translation($t$) functions are parameterized by the same fully connected neural network(FNN). More specifically, for first transition in above example, a single FNN takes $z_{1:d}^{(k)}$ as input and outputs both $s_k^a(z_{1:d}^{(k)})$ and $t_k^a(z_{1:d}^{(k)})$. Thus, the skeleton of the FNN, in terms of the size of the layers, is as $[d, H, H, 2(D - d)]$ where, $H$ denotes the size of the two hidden layers ($H$=32 for all our experiments).

The hidden layers of FNN use a leaky rectified linear unit with slope = 0.01, while the output layer uses a hyperbolic tangent for $s$ and remains linear for $t$.

**Parameter Initialization:** We initialize the parameters of the neural networks from normal distribution $\mathcal{N}(0, 0.001^2)$. We deliberately make this choice as it corresponds to an approximate standard normal initialization for the overall normalizing flow density. To see this, first note that the output from the initialized neural networks will approximately be 0 vectors. Now, consider the affine transformation of real-NVP: at each iteration, we scale by the exponent of $s$ and offset by $t$. Thus, the overall effect is an identity transform. As the base-distribution is fixed to a standard normal, this gives as an approximate standard normal initialization.

**Number of Parameters:** For each transition, assuming $d = D/2$, the parameters of the FNN can be calculated as $\frac{1}{2}DH + H^2 + HD + D + 2H$ where $D$ is the number latent dimensions in the model, and $H$ is the size of the two hidden layers. The first three components in the calculation corresponds to weight matrix, and the latter two take into account the bias parameters. With T coupling layers, each comprising of 2 transitions, the overall parameter size is given by $2T(\frac{3}{2}DH + H^2 + D + 2H)$. We use $T$=10 and $H$=32, while $D$ depends on the problem.

**Scaling to higher dimension models:** Real NVP based architectures scale better to higher dimensional problems as compared to Gaussians. The parameters in Gaussian scale as $\mathcal{O}(D^2)$ while they scale linearly $\mathcal{O}(D)$ for real-NVP, if we fix other parameters($T$ and $H$). However, for lower dimensional problem the number of parameters for real-NVP is more.

# F   Selection of Best model

We choose the model that achieves best average-objective, averaged over the entire optimization trace. This is different from, perhaps a more natural, final value based selection rule where one evaluates on a smaller batch of fresh samples; smaller compared to number of samples used for final metric evaluation. We found average objective to be more reliable indicator of the performance in practice. In our preliminary experiments, models selected from the maximum average-objective out-performed the ones selected based on the maximum final value; the comparison was based on the final metric value evaluated using a fresh batch of 10,000 samples.

# G   Full list of models

We present the complete list of models used in our analysis Table 1. The descriptions in the table have been manually extracted, see Stan-example model repository [35] for more details.

# H   Complete Table of results

Table 1: This table presents attributes of all the models from the Stan model library [35, 36] that have been used in this analysis. The attribute are $|z|$ = # of latent dimensions, $n$ = # of data points, and $r = \frac{\text{\# of latent dimensions}}{\text{\# of data points}}$

| Id | Model name | $|z|$ | $n$ | $r$ | Model Description |
|---|---|---|---|---|---|
| 1 | lsat | 1006 | 818 | 1.2298 | One-parameter Rasch model for LSAT student response |
| 2 | Mh | 388 | 385 | 1.0078 | Heterogeneity model for closed population size estimation from capture-recapture data with individual effects |
| 3 | test_simplex | 3 | 10 | 0.3000 | Simplex estimator |
| 4 | endo3 | 184 | 626 | 0.2939 | Conditional inference model for case-control study on endometrial cancer |
| 5 | gp_predict | 265 | 989 | 0.2679 | Model for predicting out-of-sample observations by fitting hyperparams of a latent variable Gaussian process with exponentiated quadratic kernel and Gaussian likelihood |
| 6 | Mth | 394 | 1935 | 0.2036 | Combined model for for closed population size estimation with both, time and individual effects |
| 7 | oxford | 244 | 1226 | 0.1990 | A mixture model for the log odds ratio to analyze Oxford childhood cancer data |
| 8 | cjs_mnl | 22 | 132 | 0.1667 | CJS model for capture-recapture problem with multinomial likelihood |
| 9 | hepatitis | 218 | 1596 | 0.1366 | Normal hierarchical model with measurement error in Hb titre in children post Hepatitis vaccination |
| 10 | normal_multi | 100 | 826 | 0.1211 | Basic Multi-variate estimators for normal and student-t distributions |
| 11 | hiv_chr | 173 | 1476 | 0.1172 | Multi-level linear model with varying slope and intercept for Zinc diet experiment on HIV positive children |
| 12 | electric_1c_chr | 116 | 1248 | 0.0929 | Multi-level linear model with varying intercept and slope for the effect of exposure to television show, The Electric Company |
| 13 | electric_1a_chr | 112 | 1248 | 0.0897 | Multi-level linear model with group level factors for the effect of exposure to television show, The Electric Company |
| 14 | electric_chr | 100 | 1248 | 0.0801 | Multi-level linear model with varying intercept for the effect of exposure to television show, The Electric Company |
| 15 | radon_vary_si_chr | 175 | 4595 | 0.0381 | Multi-level linear model with group level predictors to estimate radon levels. |
| 16 | lda | 33 | 1157 | 0.0285 | Latent Dirichlet Allocation |
| 17 | radon_redundant_chr | 88 | 4595 | 0.0192 | Multi-level linear model with varying intercept and redundant parameterization and the Choo-Hoffman parametrization |
| 18 | naive_bayes | 39 | 4124 | 0.0095 | Naive Bayes classifier |
| 19 | mesquite_volume | 3 | 322 | 0.0093 | Linear model with one transformed predictor and log transformation to measure the yield of mesquite bushes |
| 20 | cjs_t_t | 22 | 3960 | 0.0056 | CJS model for capture-recapture problem with parameter identifiability |
| 21 | irt_multilevel | 503 | 90671 | 0.0055 | Item response theory multi-level logistic model |
| 22 | irt | 501 | 90941 | 0.0055 | Item response theory 2-p logistic model |
| 23 | congress | 4 | 1029 | 0.0039 | Linear model to predict the 1988 election from 1986 election |
| 24 | dogs | 3 | 775 | 0.0039 | Multi-level logistic regression model for behavioral learning experiment on dogs |
| 25 | Dynocc | 29 | 7500 | 0.0039 | Dynamic (multi-season) site-occupancy Hidden Markov Model |
| 26 | multi_logit | 32 | 9842 | 0.0033 | Multinomial logistic regression |
| 27 | electric_one_pred | 3 | 1248 | 0.0024 | Lin. model with one predictor |
| 28 | election88 | 55 | 104094 | 0.0005 | Multi-level logistic regression model with group level predictors to predict Republican candidate in 1988 elections |
| 29 | wells | 2 | 15100 | 0.0001 | Generalized linear model with logit link function and one predictor to predict shift to a safer well in Bangladesh |
| 30 | wells_dist | 2 | 15100 | 0.0001 | Generalized linear model with logit link function and one predictor to predict shift to a safer well in Bangladesh |

Table 2: This table presents the results for ADVI baseline. ADVI runs with high variability in performance; our ADVI implementation diverges for at least 1 random trial for 10 out of 30 models.

| | $q_\phi$ family | Full-rank Gaussian | | |
| | Step-search scheme | ADVI | | |
| | $\nabla_\phi$ | Closed form entropy | | |
| | LI | Not Used | | |
| | IWVI $M_{training}$ | 1 | | |
| | IWVI $M_{sampling}$ | 1 | | |
| | Method from Outline | ADVI Baseline | | |
| | Independent Trial | Trial 1 | Trial 2 | Trial 3 |
| Id | Model Name | | | |
|---|---|---|---|---|
| 1 | lsat | nan | nan | nan |
| 2 | Mh | 19.9685 | 19.9831 | 19.9521 |
| 3 | test_simplex | -4.4433 | -4.4408 | -4.4373 |
| 4 | endo3 | -127.2903 | -127.3857 | -127.2958 |
| 5 | gp_predict | 300.0260 | 300.1867 | 300.0258 |
| 6 | Mth | -152.6560 | -152.6696 | -152.7633 |
| 7 | oxford | -4401.0033 | nan | -4520.5537 |
| 8 | cjs_mnl | -452.8369 | -452.6761 | -452.6022 |
| 9 | hepatitis | -54.1560 | -54.1572 | -54.1076 |
| 10 | normal_multi | -40367.4856 | -40365.4223 | -40368.5330 |
| 11 | hiv_chr | nan | -74.3163 | nan |
| 12 | electric_1c_chr | -287.4185 | -292.0925 | -286.8471 |
| 13 | electric_1a_chr | -428.0037 | -429.2815 | -437.4855 |
| 14 | electric_chr | -514.0052 | -557.4496 | -513.4389 |
| 15 | radon_vary_si_chr | -102.4855 | -102.4855 | -102.5110 |
| 16 | lda | -344.5263 | -344.5982 | -344.3087 |
| 17 | radon_redundant_chr | nan | nan | -299.6408 |
| 18 | naive_bayes | -3615.4987 | -3615.4900 | -3615.5445 |
| 19 | mesquite_volume | 12.4846 | 12.4895 | nan |
| 20 | cjs_t_t | -452.8026 | -452.6712 | -452.5898 |
| 21 | irt_multilevel | nan | nan | nan |
| 22 | irt | -15460.1601 | -15460.3696 | -15460.2664 |
| 23 | congress | 736.2241 | nan | 738.1187 |
| 24 | dogs | -298.4544 | -298.5008 | -298.3045 |
| 25 | Dynocc | -2126.6232 | -2126.5978 | -2126.6179 |
| 26 | multi_logit | -554.9885 | -554.6617 | -554.7382 |
| 27 | electric_one_pred | nan | nan | -657.4681 |
| 28 | election88 | -7555.5669 | -7561.2245 | -7556.6900 |
| 29 | wells_dist | -2274.6544 | nan | -2053.4626 |
| 30 | wells | nan | -2041.9096 | nan |

Table 3: This table provides results for method that uses the closed-form entropy gradient with our comprehensive step-search scheme. We further provide *additional* the results by using Laplace Initialization scheme and using IW-sampling at inference time.

| | | | | | | | | | | | | | |
|---|---|---|---|---|---|---|---|---|---|---|---|---|---|
| $q_\phi$ family | | Full-rank Gaussian | | | | | | | | | | | |
| Step-search scheme | | Comprehensive step-search | | | | | | | | | | | |
| $\nabla_\phi$ | | Closed form entropy | | | | | | | | | | | |
| LI | | Not Used | | | | | | Used | | | | | |
| IWVI $M_{training}$ | | 1 | | | | | | 1 | | | | | |
| IWVI $M_{sampling}$ | | 1 | | | 10 | | | 1 | | | 10 | | |
| Method from Outline | | (0) | | | Additional | | | Additional | | | Additional | | |
| Independent Trial | | Trial 1 | Trial 2 | Trial 3 | Trial 1 | Trial 2 | Trial 3 | Trial 1 | Trial 2 | Trial 3 | Trial 1 | Trial 2 | Trial 3 |
| Id | Model Name | | | | | | | | | | | | |
| 1 | lsat | -1560.3229 | -1560.2861 | -1560.2925 | -1558.4440 | -1558.3792 | -1558.4386 | -2666.4087 | -2667.7454 | -2667.5157 | -2621.9313 | -2622.5349 | -2622.8303 |
| 2 | Mh | 19.5196 | 19.5469 | 19.5496 | 20.4490 | 20.4554 | 20.4710 | 19.0324 | 19.0649 | 19.0763 | 20.2788 | 20.2878 | 20.2921 |
| 3 | test_simplex | -4.4463 | -4.4483 | -4.4402 | -4.3669 | -4.3745 | -4.3603 | -4.4438 | -4.4453 | -4.4496 | -4.3673 | -4.3731 | -4.3702 |
| 4 | endo3 | -127.5089 | -127.5777 | -127.5220 | -121.1431 | -121.4293 | -121.2687 | -127.5829 | -127.6264 | -127.5953 | -121.3321 | -121.2752 | -121.4067 |
| 5 | gp_predict | 198.4425 | 202.6357 | 197.5704 | 238.4563 | 238.6492 | 235.7607 | 300.0484 | 299.9787 | 300.0156 | 300.9871 | 300.9911 | 301.0216 |
| 6 | Mth | -153.0663 | -153.0603 | -153.0812 | -152.3153 | -152.2698 | -152.3311 | -153.3603 | -153.3579 | -153.3422 | -152.4340 | -152.4214 | -152.4212 |
| 7 | oxford | -4333.4364 | -4333.9387 | -4333.7062 | -4331.4304 | -4331.5858 | -4331.5967 | -4.1115e+37 | -5.9831e+44 | -4.0048e+42 | -1.1556e+05 | -3.3595e+05 | -62689.8962 |
| 8 | cjs_mnl | -452.6434 | -452.6689 | -452.6537 | -452.0531 | -452.0837 | -452.0648 | -452.6118 | -452.6235 | -452.6506 | -452.0388 | -452.0014 | -452.0169 |
| 9 | hepatitis | -54.6239 | -54.6593 | -54.6598 | -53.4851 | -53.4839 | -53.5132 | -161.6031 | -161.7103 | -161.4679 | -156.3505 | -156.4540 | -156.2607 |
| 10 | normal_multi | -74650.2692 | -74683.3277 | -74649.8184 | -73234.4113 | -72785.9847 | -72816.0867 | -16918.5887 | -16918.5877 | -16918.5881 | -16918.5867 | -16918.5857 | -16918.5862 |
| 11 | hiv_chr | -74.8046 | -74.7951 | -74.7993 | -73.6099 | -73.5665 | -73.6207 | -74.8014 | -74.8173 | -74.8080 | -73.6571 | -73.6500 | -73.6638 |
| 12 | electric_1c_chr | -285.7230 | -285.8559 | -286.4202 | -281.1185 | -281.2912 | -281.4158 | -285.6876 | -285.9342 | -286.3901 | -281.1429 | -281.3092 | -281.3765 |
| 13 | electric_1a_chr | -424.5786 | -424.5153 | -424.4387 | -422.3741 | -422.2738 | -422.1691 | -424.5125 | -424.4337 | -424.5577 | -422.3750 | -422.2143 | -422.3517 |
| 14 | electric_chr | -512.3681 | -512.2423 | -512.2471 | -511.2007 | -511.0951 | -511.0922 | -512.3493 | -512.2394 | -512.2689 | -511.2062 | -511.0864 | -511.0979 |
| 15 | radon_vary_si_chr | -102.8430 | -102.8273 | -102.8068 | -101.9041 | -101.8598 | -101.8806 | -102.8266 | -102.8438 | -102.8082 | -101.8498 | -101.8824 | -101.8779 |
| 16 | lda | -344.2642 | -344.3194 | -344.2521 | -342.7178 | -342.7799 | -342.7485 | -344.2697 | -344.2932 | -344.2646 | -342.7635 | -342.7396 | -342.7490 |
| 17 | radon_redundant_chr | -223.6925 | -225.3442 | -221.3751 | -218.1183 | -217.9346 | -217.5981 | -223.5919 | -467.7711 | -467.6689 | -218.0208 | -461.9628 | -461.9314 |
| 18 | naive_bayes | -3615.4642 | -3615.4686 | -3615.4654 | -3615.1421 | -3615.1434 | -3615.1403 | -3615.3055 | -3615.3207 | -3615.3055 | -3615.1060 | -3615.1259 | -3615.1006 |
| 19 | mesquite_volume | 12.4224 | 12.3813 | 12.3800 | 12.4977 | 12.5056 | 12.4976 | 12.4851 | 12.4834 | 12.4865 | 12.5077 | 12.5096 | 12.5132 |
| 20 | cjs_t_t | -452.6244 | -452.6379 | -452.6369 | -452.0433 | -452.0053 | -452.0335 | -452.6478 | -452.6525 | -452.6423 | -452.0485 | -452.0301 | -452.0321 |
| 21 | irt_multilevel | -14611.3903 | -14611.3943 | -14611.3818 | -14609.1618 | -14609.1189 | -14609.0938 | -14608.9681 | -14608.9780 | -14608.9549 | -14608.1255 | -14608.1816 | -14608.1382 |
| 22 | irt | -15427.1794 | -15427.2162 | -15427.1530 | -15424.9416 | -15424.9849 | -15424.8847 | -15424.2319 | -15424.2349 | -15424.2291 | -15423.8601 | -15423.8488 | -15423.8540 |
| 23 | congress | 738.5902 | 738.4774 | 738.4840 | 738.7764 | 738.6945 | 738.7785 | 738.7884 | 738.7858 | 738.7842 | 738.7941 | 738.7970 | 738.7933 |
| 24 | dogs | -298.3675 | -298.2871 | -298.3304 | -298.2795 | -298.2588 | -298.2810 | -298.2638 | -298.2646 | -298.2660 | -298.2594 | -298.2597 | -298.2593 |
| 25 | Dynocc | -2126.6299 | -2126.6223 | -2126.6431 | -2125.8526 | -2125.8764 | -2125.8708 | -2126.5057 | -2126.4791 | -2126.4745 | -2125.8125 | -2125.7992 | -2125.7765 |
| 26 | multi_logit | -553.8753 | -553.8865 | -553.8795 | -553.4833 | -553.4901 | -553.5009 | -553.5695 | -553.5774 | -553.5662 | -553.4546 | -553.4547 | -553.4509 |
| 27 | electric_one_pred | -641.6253 | -641.5721 | -641.6571 | -641.5684 | -641.5582 | -641.5811 | -641.5624 | -641.5667 | -641.5647 | -641.5572 | -641.5598 | -641.5574 |
| 28 | election88 | -7534.1590 | -7534.0445 | -7534.1257 | -7533.5235 | -7533.4941 | -7533.5037 | -7533.6499 | -7533.6635 | -7533.6472 | -7533.4165 | -7533.4340 | -7533.4334 |
| 29 | wells_dist | -2046.5482 | -2046.5790 | -2046.5165 | -2046.5134 | -2046.5195 | -2046.5087 | -2046.5399 | -2046.5277 | -2046.5413 | -2046.5147 | -2046.5119 | -2046.5111 |
| 30 | wells | -2041.9656 | -2041.9106 | -2041.9605 | -2041.9170 | -2041.9045 | -2041.9130 | -2041.9045 | -2041.9050 | -2041.9042 | -2041.9043 | -2041.9044 | -2041.9040 |

Table 4: This table provides results for Gaussian VI that uses the "full" entropy gradient from Equation (3) with our comprehensive step-search scheme. We further provide *additional* the results when using Laplace Initialization scheme and using IW-sampling at inference time.

| | | | | | | | | | | | | | |
|---|---|---|---|---|---|---|---|---|---|---|---|---|
| $q_\phi$ family | Full-rank Gaussian | | | | | | | | | | | |
| Step-search scheme | Comprehensive step-search | | | | | | | | | | | |
| $\nabla_\phi$ | Estimated without dropping score function term–full gradient | | | | | | | | | | | |
| LI | Used | | | | | | Not Used | | | | | |
| IWVI $M_{training}$ | 1 | | | | | | 1 | | | | | |
| IWVI $M_{sampling}$ | 1 | | | 10 | | | 1 | | | 10 | | |
| Method from Outline | Additional | | | Additional | | | Additional | | | Additional | | |
| Independent Trial | Trial 1 | Trial 2 | Trial 3 | Trial 1 | Trial 2 | Trial 3 | Trial 1 | Trial 2 | Trial 3 | Trial 1 | Trial 2 | Trial 3 |
| Id Model Name | | | | | | | | | | | | |
| 1 lsat | nan | nan | nan | nan | nan | nan | nan | nan | nan | nan | nan | nan |
| 2 Mh | 19.0670 | 19.1056 | 19.0700 | 20.3383 | 20.3370 | 20.3582 | 19.5593 | 19.5481 | 19.5473 | 20.4868 | 20.5185 | 20.4844 |
| 3 test_simplex | -4.4519 | -4.4444 | -4.4495 | -4.3659 | -4.3770 | -4.3691 | -4.4458 | -4.4436 | -4.4474 | -4.3599 | -4.3712 | -4.3588 |
| 4 endo3 | -127.5600 | -127.5014 | -127.5194 | -121.3365 | -121.2019 | -121.2040 | -127.4403 | -127.5594 | -127.5994 | -121.1946 | -121.2420 | -121.3132 |
| 5 gp_predict | 300.0274 | nan | 299.9987 | 300.9504 | nan | 300.9560 | 211.0246 | 195.5789 | 204.1682 | 243.6565 | 234.2550 | 241.2274 |
| 6 Mth | -153.3398 | -153.3258 | -153.3389 | -152.3910 | -152.4162 | -152.4159 | -153.0968 | -153.0545 | -153.0452 | -152.3186 | -152.2779 | -152.2451 |
| 7 oxford | -4.0948e+35 | -8.6339e+39 | -1.3128e+36 | -1.1727e+05 | -1.5272e+05 | -1.3957e+05 | -4333.4798 | -4334.0175 | -4333.8457 | -4331.4394 | -4331.5769 | -4331.6134 |
| 8 cjs_mnl | -452.6258 | -452.6614 | -452.6494 | -452.0289 | -452.0753 | -452.0496 | -452.6556 | -452.6672 | -452.6334 | -452.0531 | -452.0783 | -452.0089 |
| 9 hepatitis | -161.6162 | -161.6264 | -161.4694 | -156.4100 | -156.1979 | -156.3142 | -54.6590 | -54.6527 | -54.6491 | -53.5409 | -53.4952 | -53.5208 |
| 10 normal_multi | -16918.5889 | -16918.5874 | -16918.5888 | -16918.5869 | -16918.5853 | -16918.5869 | -74666.1985 | -74664.2850 | -74649.3550 | -73263.3554 | -72766.8331 | -72784.2473 |
| 11 hiv_chr | -74.8015 | -74.7940 | -74.8085 | -73.6664 | -73.6118 | -73.6528 | -74.7896 | -74.8136 | -74.7875 | -73.6239 | -73.6283 | -73.5881 |
| 12 electric_1c_chr | -285.7864 | -285.8934 | -286.3494 | -281.2761 | -281.3296 | -281.4412 | -285.7462 | -286.0667 | -286.4748 | -281.1279 | -281.4554 | -281.4536 |
| 13 electric_1a_chr | -424.4836 | -424.4773 | -424.5606 | -422.2672 | -422.2925 | -422.3151 | -424.4795 | -424.5070 | -424.4835 | -422.2702 | -422.2841 | -422.2535 |
| 14 electric_chr | -512.3679 | -512.2732 | -512.2296 | -511.2040 | -511.1366 | -511.0640 | -512.3744 | -512.2624 | -512.2523 | -511.1901 | -511.1049 | -511.1025 |
| 15 radon_vary_si_chr | -102.8457 | -102.8146 | -102.8164 | -101.8595 | -101.8457 | -101.8769 | -102.8437 | -102.8427 | -102.7917 | -101.9166 | -101.9149 | -101.8222 |
| 16 lda | -344.2699 | -344.2947 | -344.2570 | -342.7400 | -342.7795 | -342.7185 | -344.3098 | -344.3006 | -344.2416 | -342.7539 | -342.7678 | -342.7477 |
| 17 radon_redundant_chr | -467.9613 | -467.9049 | -467.7597 | -462.4864 | -462.3585 | -462.1381 | -223.6050 | -2935.5608 | -2948.1397 | -218.0029 | -1613.4442 | -1619.0719 |
| 18 naive_bayes | -3615.3271 | -3615.3214 | -3615.3076 | -3615.1290 | -3615.1298 | -3615.1107 | -3615.4633 | -3615.4660 | -3615.4640 | -3615.1420 | -3615.1252 | -3615.1548 |
| 19 mesquite_volume | 12.4923 | 12.4882 | 12.4795 | 12.5160 | 12.5119 | 12.5073 | 12.4174 | 12.3694 | 12.3828 | 12.4955 | 12.5047 | 12.5044 |
| 20 cjs_t_t | -452.6280 | -452.6403 | -452.6759 | -452.0359 | -452.0043 | -452.0893 | -452.6438 | -452.6591 | -452.6367 | -452.0456 | -452.0408 | -452.0151 |
| 21 irt_multilevel | -14608.9643 | -14608.9678 | -14608.9776 | -14608.1555 | -14608.1392 | -14608.1672 | -14611.3798 | -14611.3839 | -14611.3849 | -14609.1356 | -14609.0725 | -14609.1173 |
| 22 irt | -15424.2420 | -15424.2385 | -15424.2292 | -15423.8697 | -15423.8673 | -15423.8487 | -15427.2185 | -15427.1877 | -15427.1430 | -15424.9813 | -15424.9345 | -15424.9523 |
| 23 congress | 738.7869 | 738.7817 | 738.7882 | 738.7929 | 738.7934 | 738.7973 | 738.6008 | 738.4618 | 738.4788 | 738.7810 | 738.6687 | 738.7758 |
| 24 dogs | -298.2633 | -298.2652 | -298.2677 | -298.2589 | -298.2598 | -298.2612 | -298.3626 | -298.2907 | -298.3202 | -298.2743 | -298.2622 | -298.2661 |
| 25 Dynocc | -2126.4982 | -2126.4909 | -2126.5233 | -2125.7935 | -2125.7953 | -2125.8381 | -2126.6111 | -2126.6211 | -2126.5974 | -2125.8136 | -2125.8503 | -2125.8062 |
| 26 multi_logit | -553.5636 | -553.5538 | -553.5567 | -553.4501 | -553.4276 | -553.4351 | -553.8916 | -553.8986 | -553.8756 | -553.4952 | -553.5075 | -553.4867 |
| 27 electric_one_pred | -641.5643 | -641.5670 | -641.5643 | -641.5585 | -641.5605 | -641.5569 | -641.6257 | -641.5759 | -641.6568 | -641.5661 | -641.5609 | -641.5806 |
| 28 election88 | -7533.6598 | -7533.6606 | -7533.6501 | -7533.4296 | -7533.4350 | -7533.4345 | -7534.1616 | -7534.0692 | -7534.1277 | -7533.5377 | -7533.5224 | -7533.5308 |
| 29 wells_dist | -2046.5415 | -2046.5281 | -2046.5470 | -2046.5172 | -2046.5110 | -2046.5149 | -2046.5478 | -2046.5839 | -2046.5186 | -2046.5136 | -2046.5238 | -2046.5112 |
| 30 wells | -2041.9040 | -2041.9044 | -2041.9042 | -2041.9038 | -2041.9038 | -2041.9040 | -2041.9630 | -2041.9106 | -2041.9583 | -2041.9132 | -2041.9043 | -2041.9110 |

Table 5: This table provides results when using STL gradient from Equation (3) with our comprehensive step-search scheme. We further provide *additional* the results when using Laplace Initialization scheme and using IW-sampling at inference time.

| | | LI Used, $M_{sampling}$ 1, (2) | | | LI Used, $M_{sampling}$ 10, Additional | | | LI Not Used, $M_{sampling}$ 1, (1) | | | LI Not Used, $M_{sampling}$ 10, (3a) | | |
|---|---|---|---|---|---|---|---|---|---|---|---|---|---|
| Id | Model Name | Trial 1 | Trial 2 | Trial 3 | Trial 1 | Trial 2 | Trial 3 | Trial 1 | Trial 2 | Trial 3 | Trial 1 | Trial 2 | Trial 3 |
| 1 | lsat | -4091.5673 | -4091.2330 | -4092.0852 | -4056.4919 | -4056.1063 | -4055.9834 | -1558.7810 | -1558.7472 | -1558.7390 | -1557.7991 | -1557.7429 | -1557.7450 |
| 2 | Mh | 20.0356 | 20.0303 | 20.0047 | 20.6543 | 20.6089 | 20.6071 | 20.0312 | 20.0099 | 20.0382 | 20.6310 | 20.5794 | 20.6332 |
| 3 | test_simplex | -4.4462 | -4.4426 | -4.4465 | -4.3756 | -4.3697 | -4.3621 | -4.4479 | -4.4497 | -4.4429 | -4.3757 | -4.3774 | -4.3707 |
| 4 | endo3 | -127.4334 | -127.4200 | -127.4744 | -121.3711 | -121.3008 | -121.3031 | -127.4760 | -127.4425 | -127.3645 | -121.2750 | -121.3448 | -121.1234 |
| 5 | gp_predict | 300.3652 | 300.3384 | 300.3369 | 301.0796 | 301.0768 | 301.0742 | 221.0104 | 184.9239 | 199.2730 | 251.7177 | 229.4787 | 236.0091 |
| 6 | Mth | -152.5692 | -152.5807 | -152.5665 | -152.1654 | -152.1577 | -152.1657 | -152.5853 | -152.5776 | -152.5712 | -152.1531 | -152.1763 | -152.1694 |
| 7 | oxford | -4.7437e+37 | -1.6905e+39 | -6.5444e+38 | -1.0902e+05 | -63518.2033 | -2.0817e+05 | -4332.4161 | -4332.8902 | -4332.2031 | -4331.2022 | -4331.2105 | -4331.0641 |
| 8 | cjs_mnl | -452.5887 | -452.6104 | -452.6059 | -452.0214 | -452.0512 | -452.0276 | -452.5971 | -452.6071 | -452.5779 | -452.0397 | -452.0621 | -452.0186 |
| 9 | hepatitis | -158.9974 | -159.0309 | -159.2058 | -154.5824 | -154.7220 | -154.8867 | -53.7993 | -53.8159 | -53.8126 | -53.1213 | -53.1084 | -53.0708 |
| 10 | normal_multi | -16918.5862 | -16918.5866 | -16918.5863 | -16918.5861 | -16918.5860 | -16918.5862 | -74671.5739 | -74669.7508 | -74673.7012 | -73242.5592 | -72773.3522 | -72815.9822 |
| 11 | hiv_chr | -74.2120 | -160.9142 | -160.7770 | -73.4592 | -153.4229 | -153.1253 | -74.1980 | -74.2091 | -74.2213 | -73.4454 | -73.4351 | -73.4323 |
| 12 | electric_1c_chr | -284.9183 | -285.3767 | -284.2679 | -280.8849 | -280.9905 | -280.4814 | -284.8103 | -285.3329 | -284.1304 | -280.8633 | -280.9278 | -280.5198 |
| 13 | electric_1a_chr | -423.6188 | -423.5255 | -423.5899 | -421.9740 | -421.9125 | -421.8940 | -423.6309 | -423.5165 | -423.6198 | -421.9503 | -421.9002 | -421.9364 |
| 14 | electric_chr | -511.5287 | -511.4958 | -511.6702 | -510.9102 | -510.9387 | -510.9712 | -511.5600 | -511.5064 | -511.6432 | -510.9402 | -510.9655 | -510.9761 |
| 15 | radon_vary_si_chr | -211.7987 | -212.1275 | -212.0410 | -204.6177 | -204.6255 | -204.6489 | -102.3823 | -102.3710 | -102.3839 | -101.7380 | -101.6922 | -101.7524 |
| 16 | lda | -344.2253 | -344.2154 | -344.2544 | -342.7557 | -342.7557 | -342.7813 | -344.2538 | -344.2386 | -344.2659 | -342.7780 | -342.7440 | -342.8067 |
| 17 | radon_redundant_chr | -467.5724 | -557.4908 | -466.8820 | -463.1025 | -532.0482 | -462.2899 | -2924.6354 | -2934.9166 | -2954.9253 | -1563.8479 | -1611.5559 | -1629.3424 |
| 18 | naive_bayes | -3615.2914 | -3615.2818 | -3615.2835 | -3615.1175 | -3615.1048 | -3615.1145 | -3615.2889 | -3615.2737 | -3615.2853 | -3615.1151 | -3615.1053 | -3615.1042 |
| 19 | mesquite_volume | 12.4858 | 12.4884 | 12.4881 | 12.5083 | 12.5136 | 12.5123 | 12.4874 | 12.4614 | 12.4514 | 12.5127 | 12.5060 | 12.5069 |
| 20 | cjs_t_t | -452.5757 | -452.5689 | -452.6060 | -451.9996 | -452.0086 | -452.0482 | -452.5950 | -452.5798 | -452.5913 | -452.0293 | -452.0230 | -452.0237 |
| 21 | irt_multilevel | -14608.3948 | -14608.3807 | -14608.3959 | -14608.0374 | -14608.0328 | -14608.0398 | -14608.3791 | -14608.3849 | -14608.3966 | -14608.0311 | -14608.0351 | -14608.0415 |
| 22 | irt | -15424.0709 | -15424.1942 | -15424.0688 | -15423.8320 | -15423.8400 | -15423.8318 | -15424.1996 | -15424.2093 | -15424.2063 | -15423.8217 | -15423.8582 | -15423.8520 |
| 23 | congress | 738.7871 | 738.7906 | 738.7895 | 738.7920 | 738.7954 | 738.7949 | 738.7803 | 738.7863 | 738.7861 | 738.7929 | 738.7949 | 738.7942 |
| 24 | dogs | -298.2638 | -298.2635 | -298.2641 | -298.2591 | -298.2595 | -298.2590 | -298.2680 | -298.2696 | -298.2644 | -298.2587 | -298.2580 | -298.2577 |
| 25 | Dynocc | -2126.5165 | -2126.5050 | -2126.4933 | -2125.7914 | -2125.8100 | -2125.7789 | -2126.5121 | -2126.5112 | -2126.4865 | -2125.8071 | -2125.8183 | -2125.7419 |
| 26 | multi_logit | -553.5765 | -553.5475 | -553.5493 | -553.4467 | -553.4340 | -553.4499 | -553.5746 | -553.5669 | -553.5647 | -553.4531 | -553.4447 | -553.4312 |
| 27 | electric_one_pred | -641.5628 | -641.5642 | -641.5635 | -641.5576 | -641.5578 | -641.5579 | -641.5701 | -641.5706 | -641.5667 | -641.5591 | -641.5604 | -641.5600 |
| 28 | election88 | -7533.5352 | -7533.5890 | -7533.5733 | -7533.3942 | -7533.4317 | -7533.4243 | -7533.5530 | -7533.5750 | -7533.5676 | -7533.4176 | -7533.4129 | -7533.4223 |
| 29 | wells_dist | -2046.5099 | -2046.5101 | -2046.5099 | -2046.5090 | -2046.5090 | -2046.5090 | -2046.5104 | -2046.5095 | -2046.5094 | -2046.5098 | -2046.5093 | -2046.5092 |
| 30 | wells | -2041.9040 | -2041.9040 | -2041.9040 | -2041.9039 | -2041.9039 | -2041.9040 | -2041.9041 | -2041.9039 | -2041.9041 | -2041.9040 | -2041.9037 | -2041.9040 |

Table configuration header:

| $q_\phi$ family | Full-rank Gaussian |
|---|---|
| Step-search scheme | Comprehensive step-search |
| $\nabla_\phi$ | Estimate with STL |
| LI | Used / Not Used |
| IWVI $M_{training}$ | 1 / 1 |
| IWVI $M_{sampling}$ | 1 / 10 / 1 / 10 |
| Method from Outline | (2) / Additional / (1) / (3a) |

Table 6: This table provides results for importance-weighted training for Gaussian $q_\phi$ optimized with our comprehensive step-search scheme. We further provide *additional* the results when using Laplace Initialization scheme and using the regular IW-ELBO gradient of Equation (5)

| | $q_\phi$ family | Full-rank Gaussian | | | | | | | | | | | |
|---|---|---|---|---|---|---|---|---|---|---|---|---|---|
| | Step-search scheme | Comprehensive step-search | | | | | | | | | | | |
| | $\nabla_\phi$ | Estimated without dropping the score-function term | | | | | | Estimated with DReG | | | | | |
| | LI | Used | | | Not Used | | | Used | | | Not Used | | |
| | IWVI $M_{training}$ | 10 | | | 10 | | | 10 | | | 10 | | |
| | IWVI $M_{sampling}$ | 10 | | | 10 | | | 10 | | | 10 | | |
| | Method from Outline | Additional | | | Additional | | | Additional | | | (3b) | | |
| | Independent Trial | Trial 1 | Trial 2 | Trial 3 | Trial 1 | Trial 2 | Trial 3 | Trial 1 | Trial 2 | Trial 3 | Trial 1 | Trial 2 | Trial 3 |
| Id | Model Name | | | | | | | | | | | | |
| 1 | lsat | nan | nan | nan | nan | nan | nan | -3602.4668 | -3626.9508 | -3561.9721 | -1560.4725 | -1560.6102 | -1560.2753 |
| 2 | Mh | 15.4818 | 15.7472 | 15.6913 | 19.7892 | 19.7846 | 19.8196 | 19.3155 | 19.1917 | 19.1727 | 20.7624 | 20.7138 | 20.7093 |
| 3 | test_simplex | -4.3528 | -4.3557 | -4.3442 | -4.3538 | -4.3437 | -4.3491 | -4.3460 | -4.3477 | -4.3570 | -4.3503 | -4.3525 | -4.3501 |
| 4 | endo3 | -121.0991 | -121.1391 | -121.1125 | -120.9746 | -121.1293 | -121.0337 | -120.6768 | -120.6442 | -120.7764 | -120.7299 | -120.6274 | -120.7329 |
| 5 | gp_predict | nan | 300.6958 | 300.6468 | 300.7821 | 300.8882 | 300.7675 | 301.4711 | 301.4523 | 301.4776 | 301.5145 | 301.4983 | nan |
| 6 | Mth | -157.6131 | -157.8043 | -157.8328 | -152.9738 | -152.8934 | -152.9296 | -154.0925 | -154.0752 | -153.8830 | -152.0879 | -152.0961 | -152.0757 |
| 7 | oxford | -1.9933e+05 | -5.0506e+05 | -74206.7353 | -4345.0453 | -4347.0228 | -4346.2444 | -4.9035e+05 | -1.2894e+05 | -37340.6443 | -4344.8534 | -4343.0004 | -4343.6250 |
| 8 | cjs_mnl | -451.8801 | -451.8757 | -451.8600 | -451.9819 | -451.9677 | -451.9664 | -451.8670 | -451.8624 | -451.8923 | -451.8809 | -451.8810 | -451.8503 |
| 9 | hepatitis | -169.5314 | -170.4767 | -169.9543 | -69.2357 | -69.3825 | -70.9711 | -168.6359 | -168.2180 | -168.8426 | -55.9312 | -56.1853 | -56.6067 |
| 10 | normal_multi | -16918.5870 | -16918.5937 | -16918.5867 | -26750.2857 | -27498.9976 | -27460.3790 | -16918.5864 | -16918.5864 | -16918.5863 | -27788.4873 | -26875.6551 | -27865.8058 |
| 11 | hiv_chr | -176.5920 | -175.6895 | -171.8587 | -74.9593 | -74.9456 | -75.1526 | -174.1049 | -170.4290 | -170.1433 | -73.5738 | -73.5807 | -73.6530 |
| 12 | electric_1c_chr | -286.7872 | -287.3433 | -286.7932 | -315.0138 | -313.2617 | -315.6589 | -285.7240 | -286.3400 | -284.3945 | -310.0764 | -314.5847 | -311.5810 |
| 13 | electric_1a_chr | -426.9384 | -426.9405 | -428.7423 | -426.9211 | -428.3849 | -428.2209 | -422.5884 | -426.4747 | -430.4513 | -422.5316 | -422.7934 | -423.0203 |
| 14 | electric_chr | -516.2415 | -516.4791 | -516.0333 | -516.2482 | -516.6440 | -515.8745 | -515.7389 | -515.3542 | -514.7078 | -515.9188 | -515.7946 | -515.8023 |
| 15 | radon_vary_si_chr | -103.8548 | -104.1821 | -103.7281 | -102.8569 | -102.9285 | -102.9329 | -178.6936 | -181.2776 | -175.7665 | -101.6374 | -101.6283 | -101.6344 |
| 16 | lda | -342.3627 | -342.3805 | -342.3895 | -342.4098 | -342.3976 | -342.4005 | -342.3420 | -342.3486 | -342.2886 | -342.3074 | -342.3318 | -342.3201 |
| 17 | radon_redundant_chr | -356.7974 | -453.5837 | -358.3671 | -693.8210 | -693.9595 | -3186.8453 | -449.6303 | -449.8883 | -452.8245 | -671.9259 | -691.9953 | -3202.9959 |
| 18 | naive_bayes | -3615.0965 | -3615.1078 | -3615.0916 | -3615.3434 | -3615.3476 | -3615.3793 | -3615.0964 | -3615.1020 | -3615.1063 | -3615.1163 | -3615.0989 | -3615.0960 |
| 19 | mesquite_volume | 12.5129 | 12.5125 | 12.5134 | 12.3802 | 12.3888 | 12.2544 | 12.5120 | 12.5108 | 12.5123 | 12.5071 | 12.5101 | 12.5141 |
| 20 | cjs_t_t | -451.8701 | -451.8341 | -451.8730 | -452.0175 | -451.9672 | -452.0109 | -451.9038 | -451.8530 | -451.8859 | -451.8573 | -451.8625 | -451.8934 |
| 21 | irt_multilevel | -14609.2097 | -14609.3464 | -14608.4966 | -14622.5119 | -14622.2421 | -14622.6026 | -14607.9892 | -14608.0042 | -14608.0028 | -14611.7850 | -14611.8132 | -14611.5983 |
| 22 | irt | -15423.9092 | -15423.9159 | -15423.9325 | -15437.7767 | -15437.2800 | -15437.1353 | -15423.8471 | -15423.8396 | -15423.8208 | -15424.2517 | -15423.9661 | -15427.9253 |
| 23 | congress | 738.7864 | 738.7833 | 738.7801 | 738.6621 | 738.5947 | 738.6958 | 738.7922 | 738.7937 | 738.7948 | 738.7948 | 738.7948 | 738.7956 |
| 24 | dogs | -298.2718 | -298.2617 | -298.2698 | -298.4308 | -298.4959 | -298.3189 | -298.2585 | -298.2587 | -298.2598 | -298.2605 | -298.2574 | -298.2614 |
| 25 | Dynocc | -2125.6550 | -2125.6638 | -2125.7156 | -2125.8873 | -2125.9113 | -2125.8858 | -2125.6986 | -2125.6466 | -2125.5890 | -2125.6676 | -2125.6854 | -2125.6376 |
| 26 | multi_logit | -553.5210 | -553.5132 | -553.4919 | -554.1409 | -553.8148 | -554.0896 | -553.4295 | -553.4241 | -553.4322 | -553.4322 | -553.4350 | -553.4397 |
| 27 | electric_one_pred | -641.5591 | -641.5585 | -641.5581 | -641.9277 | -641.7308 | -641.6937 | -641.5579 | -641.5594 | -641.5546 | -641.5589 | -641.5605 | -641.5588 |
| 28 | election88 | -7533.4531 | -7533.4451 | -7533.4654 | -7534.4716 | -7534.6690 | -7534.5011 | -7533.4137 | -7533.4106 | -7533.4142 | -7533.4076 | -7533.4078 | -7533.4052 |
| 29 | wells_dist | -2046.5308 | -2046.5357 | -2046.5255 | -2046.6939 | -2046.5931 | -2046.5876 | -2046.5091 | -2046.5093 | -2046.5089 | -2046.5092 | -2046.5090 | -2046.5089 |
| 30 | wells | -2041.9053 | -2041.9222 | -2041.9059 | -2042.0735 | -2041.9343 | -2042.0528 | -2041.9041 | -2041.9040 | -2041.9039 | -2041.9040 | -2041.9039 | -2041.9039 |

Table 7: This table provides results for real-NVP normalizing flows optimized with our comprehensive step-search scheme with additional results from using IW-sampling.

| $q_\phi$ family | Real NVP flows | | | | | | | | | | | |
|---|---|---|---|---|---|---|---|---|---|---|---|---|
| Step-search scheme | Comprehensive step-search | | | | | | | | | | | |
| $\nabla_\phi$ | Estimated without dropping the score-function term–full gradient | | | | | | Estimated with STL | | | | | |
| LI | Not Used | | | | | | Not Used | | | | | |
| IWVI $M_{training}$ | 1 | | | | | | 1 | | | | | |
| IWVI $M_{sampling}$ | 1 | | | 10 | | | 1 | | | 10 | | |
| Method from Outline | (4a) | | | Additional | | | (4b) | | | (4c) | | |
| Independent Trial | Trial 1 | Trial 2 | Trial 3 | Trial 1 | Trial 2 | Trial 3 | Trial 1 | Trial 2 | Trial 3 | Trial 1 | Trial 2 | Trial 3 |
| Id Model Name | | | | | | | | | | | | |
| 1 lsat | -1558.0810 | -1558.0983 | -1558.0795 | -1557.4331 | -1557.4355 | -1557.4532 | -1557.7891 | -1557.7745 | -1557.7618 | -1557.3710 | -1557.3581 | -1557.3378 |
| 2 Mh | 20.6116 | 20.6099 | 20.5906 | 21.0692 | 21.0748 | 21.0498 | 20.8141 | 20.7952 | 20.7906 | 21.1431 | 21.1399 | 21.1405 |
| 3 test_simplex | nan | nan | nan | nan | nan | nan | nan | nan | nan | nan | nan | nan |
| 4 endo3 | -125.4364 | -125.6489 | -125.3696 | -119.9958 | -120.1006 | -119.8449 | -124.1439 | -124.6179 | -123.8879 | -119.0637 | -119.4145 | -118.8396 |
| 5 gp_predict | 301.0234 | 300.9385 | 133.3838 | 301.8497 | 301.8672 | 175.2945 | 291.8903 | 186.8792 | 278.3282 | 295.9186 | 222.9463 | 287.6078 |
| 6 Mth | -152.0877 | -152.1042 | -152.0981 | -151.8264 | -151.8418 | -151.8434 | -151.9116 | -151.8974 | -151.9065 | -151.7999 | -151.7864 | -151.7871 |
| 7 oxford | -4332.3919 | -4332.4422 | -4332.3601 | -4331.2495 | -4331.2409 | -4331.2086 | -4331.7142 | -4331.7552 | -4331.6724 | -4330.9652 | -4330.9303 | -4330.9149 |
| 8 cjs_mnl | -451.6828 | -451.6553 | -451.7014 | -451.5297 | -451.5138 | -451.5268 | -451.5370 | -451.5515 | -451.5312 | -451.5047 | -451.5007 | -451.5000 |
| 9 hepatitis | -52.1719 | -52.0280 | -51.9546 | -51.3164 | -51.2786 | -51.1834 | -51.1649 | -51.1504 | -51.0317 | -50.8315 | -50.7965 | -50.7713 |
| 10 normal_multi | -16937.4055 | -16924.0775 | -16940.7436 | -16930.1828 | -16920.9157 | -16930.9541 | -16919.3660 | -16933.2347 | -16919.2307 | -16918.7303 | -16926.7185 | -16918.7138 |
| 11 hiv_chr | -74.0555 | -74.4123 | -74.2517 | -73.0828 | -73.2133 | -73.1843 | -73.1723 | -73.0908 | -73.1615 | -72.8201 | -72.7956 | -72.8215 |
| 12 electric_1c_chr | -279.5105 | -279.2506 | -278.9096 | -278.1362 | -278.0216 | -277.9078 | -278.1784 | -278.8128 | -278.2789 | -277.6591 | -277.8439 | -277.6683 |
| 13 electric_1a_chr | -420.8645 | -420.9422 | -420.8577 | -420.2430 | -420.2319 | -420.2248 | -420.2447 | -420.3372 | -420.2873 | -420.0749 | -420.1021 | -420.0974 |
| 14 electric_chr | -510.9007 | -510.8399 | -511.0042 | -510.6317 | -510.6201 | -510.6661 | -510.6236 | -510.6386 | -510.6629 | -510.6001 | -510.6022 | -510.6002 |
| 15 radon_vary_si_chr | -102.3573 | -102.2121 | -102.0270 | -101.5372 | -101.5044 | -101.4406 | -101.4443 | -101.5437 | -101.4984 | -101.3241 | -101.3366 | -101.3272 |
| 16 lda | nan | -343.8487 | nan | nan | -342.5254 | nan | nan | nan | nan | nan | nan | nan |
| 17 radon_redundant_chr | -217.3711 | -217.2081 | -217.2531 | -216.9475 | -216.9049 | -216.9103 | -216.9012 | -217.0216 | -216.9071 | -216.8625 | -216.8828 | -216.8627 |
| 18 naive_bayes | -3615.3514 | -3615.2491 | -3615.3346 | -3615.1268 | -3615.1000 | -3615.1178 | -3615.0951 | -3615.1031 | -3615.1061 | -3615.0718 | -3615.0772 | -3615.0768 |
| 19 mesquite_volume | 12.4818 | 12.4943 | 12.4949 | 12.5128 | 12.5150 | 12.5175 | 12.5073 | 12.4761 | 12.5132 | 12.5085 | 12.5094 | 12.5139 |
| 20 cjs_t_t | -451.6751 | -451.6590 | -451.6910 | -451.5173 | -451.5230 | -451.5276 | -451.5337 | -451.5676 | -451.5380 | -451.5010 | -451.5019 | -451.5029 |
| 21 irt_multilevel | -14609.0529 | -14609.1944 | -14609.0893 | -14608.1901 | -14608.2447 | -14608.2274 | -14608.3838 | -14608.3419 | -14608.3424 | -14608.0398 | -14608.0113 | -14608.0313 |
| 22 irt | -15425.1807 | -15425.0129 | -15424.9788 | -15424.1654 | -15424.0792 | -15424.0399 | -15424.3563 | -15424.2689 | -15424.2745 | -15423.8821 | -15423.8861 | -15423.8640 |
| 23 congress | 738.7674 | 738.6810 | 738.4917 | 738.7953 | 738.7737 | 738.7619 | 738.6337 | 738.7920 | 738.5114 | 738.7762 | 738.7957 | 738.7648 |
| 24 dogs | -298.3447 | -303.0714 | -298.2653 | -298.2672 | -302.1189 | -298.2595 | -298.2593 | -298.2587 | -298.2587 | -298.2587 | -298.2580 | -298.2578 |
| 25 Dynocc | -2125.4343 | -2125.4809 | -2125.4959 | -2125.1858 | -2125.2057 | -2125.2117 | -2125.2266 | -2125.2430 | -2125.2304 | -2125.1621 | -2125.1556 | -2125.1584 |
| 26 multi_logit | -554.3855 | -554.2881 | -554.4112 | -553.6644 | -553.6392 | -553.6909 | -553.8005 | -553.8478 | -553.8052 | -553.4846 | -553.5251 | -553.4879 |
| 27 electric_one_pred | -641.6955 | -641.5924 | -641.7577 | -641.5716 | -641.5595 | -641.5836 | -641.5577 | -641.5823 | -641.5729 | -641.5568 | -641.5594 | -641.5580 |
| 28 election88 | -7533.5399 | -7533.5842 | -7533.6034 | -7533.4097 | -7533.4182 | -7533.4205 | -7533.4139 | -7533.3991 | -7533.4044 | -7533.3948 | -7533.3908 | -7533.3919 |
| 29 wells_dist | -14358.7734 | -2053.1972 | -2066.2398 | -2107.7104 | -2052.5297 | -2052.9980 | -2046.5242 | -2046.5117 | -2046.5132 | -2046.5109 | -2046.5098 | -2046.5107 |
| 30 wells | -2053.2450 | -2128.8088 | -2043.5417 | -2052.6375 | -2057.4116 | -2043.0341 | -2041.9040 | -2041.9045 | -2041.9038 | -2041.9039 | -2041.9043 | -2041.9037 |

Table 8: This table provides results for importance weighted training for real-NVP normalizing flows optimized with our comprehensive step-search scheme with additional results from using regular IW-ELBO gradient.

| | $q_\phi$ family | Real NVP flows | | | | | |
|---|---|---|---|---|---|---|---|
| | Step-search scheme | Comprehensive step-search | | | | | |
| | $\nabla_\phi$ | Estimated w/o dropping score-function term | | | Estimated with DReG | | |
| | LI | Not Used | | | Not Used | | |
| | IWVI $M_{training}$ | 10 | | | 10 | | |
| | IWVI $M_{sampling}$ | 10 | | | 10 | | |
| | Method from Outline | Additional | | | (4d) | | |
| | Independent Trial | Trial 1 | Trial 2 | Trial 3 | Trial 1 | Trial 2 | Trial 3 |
| Id | Model Name | | | | | | |
| 1 | lsat | -1557.7324 | -1557.7093 | -1557.7707 | -1557.3724 | -1557.3555 | -1557.3413 |
| 2 | Mh | 20.8262 | 20.8106 | 20.8031 | 21.1380 | 21.1331 | 21.1265 |
| 3 | test_simplex | nan | nan | nan | nan | nan | nan |
| 4 | endo3 | -120.3320 | -120.4250 | -120.4874 | -120.0272 | -119.9676 | -120.1827 |
| 5 | gp_predict | 300.1498 | 300.0910 | 300.6795 | 301.7336 | 301.7699 | 301.7164 |
| 6 | Mth | -151.9868 | -152.0086 | -152.0120 | -151.8052 | -151.7779 | -151.8100 |
| 7 | oxford | -4332.1323 | -4332.3033 | -4332.4134 | -4331.0128 | -4331.0230 | -4330.9522 |
| 8 | cjs_mnl | -451.7348 | -451.7418 | -451.7667 | -451.5060 | -451.5204 | -451.5041 |
| 9 | hepatitis | -52.2705 | -52.4800 | -52.5429 | -50.8712 | -51.2069 | -50.8934 |
| 10 | normal_multi | -16933.1393 | -16940.4390 | -16935.3070 | -16920.0900 | -16922.8421 | -16937.6249 |
| 11 | hiv_chr | -74.9543 | -75.1996 | -75.2645 | -73.0401 | -72.9434 | -72.8453 |
| 12 | electric_1c_chr | -279.7025 | -279.6760 | -279.8671 | -278.0520 | -278.2926 | -278.3740 |
| 13 | electric_1a_chr | -421.0590 | -421.0157 | -421.1706 | -420.1473 | -420.1479 | -420.1098 |
| 14 | electric_chr | -510.8465 | -511.4763 | -510.9585 | -510.6169 | -510.6045 | -510.6419 |
| 15 | radon_vary_si_chr | -102.8107 | -102.1768 | -102.3642 | -101.6430 | -101.3792 | -101.3284 |
| 16 | lda | -342.2148 | -342.1463 | -341.9619 | -342.1034 | nan | -342.1726 |
| 17 | radon_redundant_chr | -217.5033 | -217.3719 | -217.3199 | -219.1729 | -216.8761 | -216.8774 |
| 18 | naive_bayes | -3615.2138 | -3615.1675 | -3615.2120 | -3615.0806 | -3615.0806 | -3615.0849 |
| 19 | mesquite_volume | 12.4558 | 12.4775 | 12.4488 | 12.4897 | 12.5151 | 12.5162 |
| 20 | cjs_t_t | -451.6491 | -451.6962 | -451.7152 | -451.7311 | -451.5933 | -451.5014 |
| 21 | irt_multilevel | -14609.2872 | -14609.3237 | -14609.3317 | -14608.0970 | -14608.0508 | -14608.0362 |
| 22 | irt | -15424.9266 | -15425.1680 | -15424.9245 | -15423.8881 | -15423.9093 | -15423.9310 |
| 23 | congress | 738.7386 | 738.7660 | 738.7162 | 738.7869 | 738.7954 | 738.7907 |
| 24 | dogs | -298.3505 | -298.2761 | -298.2739 | -298.2580 | -298.2586 | -298.2581 |
| 25 | Dynocc | -2125.3477 | -2125.3606 | -2125.3213 | -2125.1688 | -2125.1674 | -2125.1792 |
| 26 | multi_logit | -555.2918 | -556.6276 | -555.3734 | -556.2806 | -553.5680 | -554.7593 |
| 27 | electric_one_pred | -641.6248 | -641.5941 | -641.6119 | -641.5573 | -641.5567 | -641.5628 |
| 28 | election88 | -7533.5460 | -7533.6522 | -7533.6817 | -7533.3927 | -7533.4052 | -7533.3914 |
| 29 | wells_dist | -2046.7589 | -2057.7702 | -2046.6208 | -2046.5093 | -2046.5150 | -2046.5097 |
| 30 | wells | -2047.6023 | -2041.9124 | -2041.9151 | -2041.9042 | -2041.9042 | -2041.9038 |

Table 9: This table presents the results for additional Diagonal Gaussian experiments. Please refer to Figure 11 and appendix B for more details.

| $q_\phi$ family | Diagonal Gaussian | | | | | | | | |
|---|---|---|---|---|---|---|---|---|---|
| Step-search scheme | Comprehensive step-search | | | | | | | | |
| $\nabla_\phi$ | Closed form entropy | | | Estimated with STL | | | | | |
| LI | Not Used | | | Not Used | | | | | |
| IWVI $M_{training}$ | 1 | | | 1 | | | | | |
| IWVI $M_{sampling}$ | 1 | | | 1 | | | 10 | | |
| Method from Outline | Additional | | | Additional | | | Additional | | |
| Independent Trial | Trial 1 | Trial 2 | Trial 3 | Trial 1 | Trial 2 | Trial 3 | Trial 1 | Trial 2 | Trial 3 |

| Id | Model Name | Trial 1 | Trial 2 | Trial 3 | Trial 1 | Trial 2 | Trial 3 | Trial 1 | Trial 2 | Trial 3 |
|---|---|---|---|---|---|---|---|---|---|---|
| 1 | lsat | -1593.5322 | -1593.4869 | -1593.3559 | -1592.8262 | -1592.4748 | -1592.5821 | -1571.0762 | -1570.9308 | -1570.8328 |
| 2 | Mh | 19.1775 | 19.1667 | 19.1541 | 19.1718 | 19.1684 | 19.1918 | 19.9559 | 19.9322 | 19.9619 |
| 3 | test_simplex | -4.4521 | -4.4428 | -4.4479 | -4.4365 | -4.4451 | -4.4432 | -4.3572 | -4.3665 | -4.3664 |
| 4 | endo3 | -128.4125 | -128.4300 | -128.4104 | -128.3513 | -128.4653 | -128.3904 | -122.0453 | -122.1725 | -122.1298 |
| 5 | gp_predict | 125.4957 | 124.4540 | 109.7607 | 119.1200 | 113.7045 | 126.2959 | 173.3857 | 170.0907 | 178.5877 |
| 6 | Mth | -153.1179 | -153.1375 | -153.1227 | -153.1145 | -153.0937 | -153.1330 | -152.5015 | -152.5002 | -152.5632 |
| 7 | oxford | -4351.8876 | -4352.1289 | -4352.0302 | -4351.8651 | -4351.9882 | -4352.0168 | -4334.3600 | -4334.3489 | -4334.3732 |
| 8 | cjs_mnl | -455.1820 | -455.1343 | -455.1980 | -455.1473 | -455.1559 | -455.1645 | -453.5222 | -453.4747 | -453.5459 |
| 9 | hepatitis | -56.9566 | -56.9034 | -56.9102 | -56.7279 | -56.7292 | -56.7131 | -55.1387 | -55.1739 | -55.1519 |
| 10 | normal_multi | -74651.8168 | -74653.5847 | -74651.7663 | -74651.0597 | -74657.3869 | -74653.1409 | -74006.7030 | -74007.7665 | -73999.5017 |
| 11 | hiv_chr | -88.0622 | -88.1118 | -88.0928 | -88.0838 | -88.1383 | -88.1089 | -86.3650 | -86.4350 | -86.3553 |
| 12 | electric_1c_chr | -293.8758 | -293.8000 | -293.7533 | -293.8438 | -293.7498 | -293.8464 | -289.9068 | -289.8101 | -289.8923 |
| 13 | electric_1a_chr | -427.1647 | -427.1551 | -427.1726 | -427.1397 | -427.1230 | -427.1873 | -424.2745 | -424.2433 | -424.3046 |
| 14 | electric_chr | -513.1907 | -513.1651 | -513.1779 | -513.1541 | -513.1480 | -513.1572 | -512.1036 | -512.1022 | -512.0917 |
| 15 | radon_vary_si_chr | -105.8260 | -105.8688 | -105.8414 | -105.8406 | -105.8733 | -105.8842 | -103.5692 | -103.6337 | -103.6410 |
| 16 | lda | -344.4616 | -344.4580 | -344.4734 | -344.4943 | -344.5078 | -344.4530 | -342.8903 | -342.8522 | -342.8561 |
| 17 | radon_redundant_chr | -218.1631 | -218.2137 | -218.2104 | -218.1262 | -218.2738 | -218.1381 | -217.4305 | -217.4860 | -217.5006 |
| 18 | naive_bayes | -3615.2682 | -3615.2812 | -3615.2838 | -3615.2553 | -3615.2565 | -3615.2691 | -3615.1094 | -3615.1041 | -3615.1181 |
| 19 | mesquite_volume | 12.1762 | 12.1763 | 12.1197 | 12.1633 | 12.1528 | 12.1737 | 12.4216 | 12.4033 | 12.4035 |
| 20 | cjs_t_t | -455.1950 | -455.1641 | -455.2009 | -455.1728 | -455.1777 | -455.1742 | -453.4716 | -453.5388 | -453.5097 |
| 21 | irt_multilevel | -14631.4012 | -14632.0799 | -14631.5253 | -14631.2708 | -14631.9146 | -14631.2243 | -14611.8698 | -14611.8590 | -14611.7534 |
| 22 | irt | -15443.3924 | -15443.1456 | -15443.8044 | -15443.3331 | -15443.2550 | -15443.4153 | -15426.3968 | -15426.4443 | -15426.4066 |
| 23 | congress | 737.0282 | 736.7399 | 736.9055 | 737.0364 | 736.8671 | 736.7321 | 737.5111 | 737.4481 | 737.5068 |
| 24 | dogs | -299.3324 | -299.3696 | -299.3739 | -299.3697 | -299.4016 | -299.3477 | -298.7402 | -298.7262 | -298.7594 |
| 25 | Dynocc | -2129.9383 | -2129.9050 | -2129.8542 | -2129.8750 | -2129.8567 | -2129.9244 | -2127.8466 | -2127.7597 | -2127.8208 |
| 26 | multi_logit | -575.8037 | -575.8263 | -575.8159 | -575.8136 | -575.7877 | -575.8724 | -572.5463 | -572.5639 | -572.6479 |
| 27 | electric_one_pred | -641.9286 | -641.9152 | -641.9057 | -641.9139 | -641.9218 | -641.9175 | -641.6821 | -641.6903 | -641.6910 |
| 28 | election88 | -7534.3065 | -7534.3075 | -7534.3145 | -7534.2861 | -7534.2890 | -7534.2778 | -7533.7389 | -7533.7166 | -7533.7257 |
| 29 | wells_dist | -2047.2697 | -2047.0170 | -2048.8710 | -2047.2355 | -2047.0116 | -2047.2456 | -2046.7341 | -2046.7397 | -2046.7401 |
| 30 | wells | -2042.4365 | -2042.4200 | -2042.3901 | -2042.4154 | -2042.4080 | -2042.4133 | -2042.1239 | -2042.1286 | -2042.0731 |

# I Complete table for per iteration training times

For completeness, we include the per iterations training times of all the VI methods we experiment with. However, these training times should be read into with caution. We interface with Pystan and Autograd for our work; this creates an extra overhead with can dominate the run-times when the models are expensive to evaluate. Further, each training instance is run on a single CPU core.

Table 10: This table presents the per iteration training times for ADVI baseline. Please refer to Table 2 for lower-bound results.

| | $q_\phi$ family | Full-rank Gaussian | | |
| | Step-search scheme | ADVI | | |
| | $\nabla_\phi$ | Closed form entropy | | |
| | LI | Not Used | | |
| | IWVI M$_{training}$ | 1 | | |
| | IWVI M$_{sampling}$ | 1 | | |
| | Independent Trial | Trial 1 | Trial 2 | Trial 3 |
| Id | Model Name | | | |
|---|---|---|---|---|
| 1 | lsat | 0.2802 | 0.2805 | 0.2800 |
| 2 | Mh | 0.1252 | 0.1294 | 0.1234 |
| 3 | test_simplex | 0.0158 | 0.0169 | 0.0154 |
| 4 | endo3 | 0.0450 | 0.0453 | 0.0458 |
| 5 | gp_predict | 1.3203 | 1.2598 | 1.2821 |
| 6 | Mth | 0.1207 | 0.1061 | 0.1069 |
| 7 | oxford | 0.0585 | 0.0582 | 0.0581 |
| 8 | cjs_mnl | 0.0242 | 0.0206 | 0.0236 |
| 9 | hepatitis | 0.0451 | 0.0456 | 0.0455 |
| 10 | normal_multi | 0.2398 | 0.2392 | 0.2391 |
| 11 | hiv_chr | 0.0422 | 0.0426 | 0.0429 |
| 12 | electric_1c_chr | 0.0256 | 0.0272 | 0.0254 |
| 13 | electric_1a_chr | 0.0361 | 0.0391 | 0.0372 |
| 14 | electric_chr | 0.0243 | 0.0262 | 0.0254 |
| 15 | radon_vary_si_chr | 0.0477 | 0.0476 | 0.0475 |
| 16 | lda | 0.0447 | 0.0518 | 0.0510 |
| 17 | radon_redundant_chr | 0.0235 | 0.0235 | 0.0236 |
| 18 | naive_bayes | 0.1284 | 0.1280 | 0.1282 |
| 19 | mesquite_volume | 0.0163 | 0.0162 | 0.0163 |
| 20 | cjs_t_t | 0.1104 | 0.1102 | 0.1044 |
| 21 | irt_multilevel | 0.7842 | 0.7865 | 0.7845 |
| 22 | irt | 0.8777 | 0.8527 | 0.8809 |
| 23 | congress | 0.0223 | 0.0196 | 0.0224 |
| 24 | dogs | 0.0440 | 0.0440 | 0.0440 |
| 25 | Dynocc | 0.1484 | 0.1423 | 0.1420 |
| 26 | multi_logit | 0.1109 | 0.1070 | 0.1076 |
| 27 | electric_one_pred | 0.0179 | 0.0179 | 0.0178 |
| 28 | election88 | 0.3556 | 0.3510 | 0.3512 |
| 29 | wells_dist | 0.0768 | 0.0769 | 0.0792 |
| 30 | wells | 0.0914 | 0.0932 | 0.0827 |

Table 11: This table provides the per iteration training times for method that uses the closed-form entropy gradient with our comprehensive step-search scheme. Refer to Table 3 for lower-bound results.

| | $q_\phi$ family | Full-rank Gaussian | | | | | | | | | | | |
| --- | --- | --- | --- | --- | --- | --- | --- | --- | --- | --- | --- | --- | --- |
| | Step-search scheme | Comprehensive step-search | | | | | | | | | | | |
| | $\nabla_\phi$ | Closed form entropy | | | | | | | | | | | |
| | LI | Not Used | | | | | | Used | | | | | |
| | IWVI $M_{training}$ | 1 | | | | | | 1 | | | | | |
| | IWVI $M_{sampling}$ | 1 | | | 10 | | | 1 | | | 10 | | |
| | Independent Trial | Trial 1 | Trial 2 | Trial 3 | Trial 1 | Trial 2 | Trial 3 | Trial 1 | Trial 2 | Trial 3 | Trial 1 | Trial 2 | Trial 3 |
| Id | Model Name | | | | | | | | | | | | |
| 1 | lsat | 0.3344 | 0.1938 | 0.3348 | 0.3344 | 0.1938 | 0.3348 | 0.3297 | 0.3278 | 0.3299 | 0.3297 | 0.3278 | 0.3299 |
| 2 | Mh | 0.0834 | 0.0949 | 0.0954 | 0.0834 | 0.0949 | 0.0954 | 0.0939 | 0.0925 | 0.0926 | 0.0939 | 0.0925 | 0.0926 |
| 3 | test_simplex | 0.0215 | 0.0212 | 0.0223 | 0.0215 | 0.0212 | 0.0223 | 0.0191 | 0.0208 | 0.0212 | 0.0191 | 0.0208 | 0.0212 |
| 4 | endo3 | 0.0292 | 0.0290 | 0.0290 | 0.0292 | 0.0290 | 0.0290 | 0.0291 | 0.0261 | 0.0291 | 0.0291 | 0.0261 | 0.0291 |
| 5 | gp_predict | 1.1382 | 1.1475 | 1.1472 | 1.1382 | 1.1475 | 1.1472 | 1.0495 | 1.0156 | 0.9774 | 1.0495 | 1.0156 | 0.9774 |
| 6 | Mth | 0.0898 | 0.0890 | 0.0890 | 0.0898 | 0.0890 | 0.0890 | 0.0876 | 0.0924 | 0.0881 | 0.0876 | 0.0924 | 0.0881 |
| 7 | oxford | 0.0340 | 0.0321 | 0.0319 | 0.0340 | 0.0321 | 0.0319 | 0.0365 | 0.0304 | 0.0366 | 0.0365 | 0.0304 | 0.0366 |
| 8 | cjs_mnl | 0.0277 | 0.0259 | 0.0281 | 0.0277 | 0.0259 | 0.0281 | 0.0210 | 0.0217 | 0.0217 | 0.0210 | 0.0217 | 0.0217 |
| 9 | hepatitis | 0.0315 | 0.0284 | 0.0282 | 0.0315 | 0.0284 | 0.0282 | 0.0299 | 0.0277 | 0.0279 | 0.0299 | 0.0277 | 0.0279 |
| 10 | normal_multi | 0.1459 | 0.1441 | 0.1450 | 0.1459 | 0.1441 | 0.1450 | 0.1350 | 0.1356 | 0.1346 | 0.1350 | 0.1356 | 0.1346 |
| 11 | hiv_chr | 0.0282 | 0.0279 | 0.0284 | 0.0282 | 0.0279 | 0.0284 | 0.0277 | 0.0304 | 0.0280 | 0.0277 | 0.0304 | 0.0280 |
| 12 | electric_1c_chr | 0.0263 | 0.0244 | 0.0262 | 0.0263 | 0.0244 | 0.0262 | 0.0246 | 0.0261 | 0.0245 | 0.0246 | 0.0261 | 0.0245 |
| 13 | electric_1a_chr | 0.0250 | 0.0247 | 0.0249 | 0.0250 | 0.0247 | 0.0249 | 0.0244 | 0.0353 | 0.0242 | 0.0244 | 0.0353 | 0.0242 |
| 14 | electric_chr | 0.0215 | 0.0215 | 0.0215 | 0.0215 | 0.0215 | 0.0215 | 0.0218 | 0.0218 | 0.0217 | 0.0218 | 0.0218 | 0.0217 |
| 15 | radon_vary_si_chr | 0.0292 | 0.0293 | 0.0307 | 0.0292 | 0.0293 | 0.0307 | 0.0259 | 0.0309 | 0.0309 | 0.0259 | 0.0309 | 0.0309 |
| 16 | lda | 0.0387 | 0.0398 | 0.0400 | 0.0387 | 0.0398 | 0.0400 | 0.0398 | 0.0356 | 0.0355 | 0.0398 | 0.0356 | 0.0355 |
| 17 | radon_redundant_chr | 0.0221 | 0.0211 | 0.0213 | 0.0221 | 0.0211 | 0.0213 | 0.0213 | 0.0219 | 0.0192 | 0.0213 | 0.0219 | 0.0192 |
| 18 | naive_bayes | 0.0734 | 0.0757 | 0.0732 | 0.0734 | 0.0757 | 0.0732 | 0.0844 | 0.0820 | 0.0820 | 0.0844 | 0.0820 | 0.0820 |
| 19 | mesquite_volume | 0.0141 | 0.0145 | 0.0142 | 0.0141 | 0.0145 | 0.0142 | 0.0145 | 0.0144 | 0.0145 | 0.0145 | 0.0144 | 0.0145 |
| 20 | cjs_t_t | 0.0679 | 0.0852 | 0.0850 | 0.0679 | 0.0852 | 0.0850 | 0.0852 | 0.0748 | 0.0755 | 0.0852 | 0.0748 | 0.0755 |
| 21 | irt_multilevel | 0.7569 | 0.7563 | 0.7526 | 0.7569 | 0.7563 | 0.7526 | 0.9201 | 0.9183 | 0.9202 | 0.9201 | 0.9183 | 0.9202 |
| 22 | irt | 0.9308 | 0.9239 | 0.9264 | 0.9308 | 0.9239 | 0.9264 | 0.9238 | 0.8785 | 0.5136 | 0.9238 | 0.8785 | 0.5136 |
| 23 | congress | 0.0177 | 0.0178 | 0.0177 | 0.0177 | 0.0178 | 0.0177 | 0.0176 | 0.0176 | 0.0175 | 0.0176 | 0.0176 | 0.0175 |
| 24 | dogs | 0.0314 | 0.0338 | 0.0316 | 0.0314 | 0.0338 | 0.0316 | 0.0357 | 0.0356 | 0.0355 | 0.0357 | 0.0356 | 0.0355 |
| 25 | Dynocc | 0.1006 | 0.1156 | 0.1011 | 0.1006 | 0.1156 | 0.1011 | 0.1165 | 0.0962 | 0.1161 | 0.1165 | 0.0962 | 0.1161 |
| 26 | multi_logit | 0.0792 | 0.0792 | 0.0789 | 0.0792 | 0.0792 | 0.0789 | 0.0819 | 0.0819 | 0.0819 | 0.0819 | 0.0819 | 0.0819 |
| 27 | electric_one_pred | 0.0229 | 0.0197 | 0.0197 | 0.0229 | 0.0197 | 0.0197 | 0.0220 | 0.0181 | 0.0223 | 0.0220 | 0.0181 | 0.0223 |
| 28 | election88 | 0.2555 | 0.2544 | 0.2548 | 0.2555 | 0.2544 | 0.2548 | 0.2474 | 0.2267 | 0.2471 | 0.2474 | 0.2267 | 0.2471 |
| 29 | wells_dist | 0.0557 | 0.0558 | 0.0558 | 0.0557 | 0.0558 | 0.0558 | 0.0578 | 0.0774 | 0.0572 | 0.0578 | 0.0774 | 0.0572 |
| 30 | wells | 0.0672 | 0.0660 | 0.0662 | 0.0672 | 0.0660 | 0.0662 | 0.0667 | 0.0655 | 0.0658 | 0.0667 | 0.0655 | 0.0658 |

Table 12: This table provides per iteration training times for Gaussian VI that uses the "full" entropy gradient from Equation (3) with our comprehensive step-search scheme. Refer to Table 4 for lower-bound results.

| | $q_\phi$ family | Full-rank Gaussian | | | | | | | | | | | |
|---|---|---|---|---|---|---|---|---|---|---|---|---|---|
| | Step-search scheme | Comprehensive step-search | | | | | | | | | | | |
| | $\nabla_\phi$ | Estimated without dropping score function term–full gradient | | | | | | | | | | | |
| | LI | Used | | | | | | Not Used | | | | | |
| | IWVI $M_{training}$ | 1 | | | | | | 1 | | | | | |
| | IWVI $M_{sampling}$ | 1 | | | 10 | | | 1 | | | 10 | | |
| | Independent Trial | Trial 1 | Trial 2 | Trial 3 | Trial 1 | Trial 2 | Trial 3 | Trial 1 | Trial 2 | Trial 3 | Trial 1 | Trial 2 | Trial 3 |
| Id | Model Name | | | | | | | | | | | | |
| 1 | lsat | nan | nan | nan | nan | nan | nan | nan | nan | nan | nan | nan | nan |
| 2 | Mh | 0.2940 | 0.3405 | 0.3059 | 0.2940 | 0.3405 | 0.3059 | 0.3256 | 0.3022 | 0.3277 | 0.3256 | 0.3022 | 0.3277 |
| 3 | test_simplex | 0.0188 | 0.0211 | 0.0208 | 0.0188 | 0.0211 | 0.0208 | 0.0207 | 0.0188 | 0.0215 | 0.0207 | 0.0188 | 0.0215 |
| 4 | endo3 | 0.0630 | 0.0652 | 0.0634 | 0.0630 | 0.0652 | 0.0634 | 0.0630 | 0.0672 | 0.0632 | 0.0630 | 0.0672 | 0.0632 |
| 5 | gp_predict | 1.1061 | nan | 1.0685 | 1.1061 | nan | 1.0685 | 0.9109 | 1.0880 | 1.0933 | 0.9109 | 1.0880 | 1.0933 |
| 6 | Mth | 0.2856 | 0.2957 | 0.3500 | 0.2856 | 0.2957 | 0.3500 | 0.2951 | 0.3391 | 0.3362 | 0.2951 | 0.3391 | 0.3362 |
| 7 | oxford | 0.1088 | 0.1041 | 0.1273 | 0.1088 | 0.1041 | 0.1273 | 0.1109 | 0.1096 | 0.1087 | 0.1109 | 0.1096 | 0.1087 |
| 8 | cjs_mnl | 0.0272 | 0.0287 | 0.0276 | 0.0272 | 0.0287 | 0.0276 | 0.0268 | 0.0277 | 0.0286 | 0.0268 | 0.0277 | 0.0286 |
| 9 | hepatitis | 0.0931 | 0.0987 | 0.0986 | 0.0931 | 0.0987 | 0.0986 | 0.0894 | 0.0993 | 0.0990 | 0.0894 | 0.0993 | 0.0990 |
| 10 | normal_multi | 0.1769 | 0.1776 | 0.1776 | 0.1769 | 0.1776 | 0.1776 | 0.1783 | 0.1756 | 0.1772 | 0.1783 | 0.1756 | 0.1772 |
| 11 | hiv_chr | 0.0626 | 0.0632 | 0.0610 | 0.0626 | 0.0632 | 0.0610 | 0.0611 | 0.0609 | 0.0593 | 0.0611 | 0.0609 | 0.0593 |
| 12 | electric_1c_chr | 0.0424 | 0.0413 | 0.0424 | 0.0424 | 0.0413 | 0.0424 | 0.0425 | 0.0430 | 0.0427 | 0.0425 | 0.0430 | 0.0427 |
| 13 | electric_1a_chr | 0.0394 | 0.0396 | 0.0404 | 0.0394 | 0.0396 | 0.0404 | 0.0397 | 0.0409 | 0.0407 | 0.0397 | 0.0409 | 0.0407 |
| 14 | electric_chr | 0.0352 | 0.0360 | 0.0343 | 0.0352 | 0.0360 | 0.0343 | 0.0350 | 0.0357 | 0.0357 | 0.0350 | 0.0357 | 0.0357 |
| 15 | radon_vary_si_chr | 0.0642 | 0.0584 | 0.0640 | 0.0642 | 0.0584 | 0.0640 | 0.0629 | 0.0642 | 0.0650 | 0.0629 | 0.0642 | 0.0650 |
| 16 | lda | 0.0488 | 0.0492 | 0.0491 | 0.0488 | 0.0492 | 0.0491 | 0.0489 | 0.0486 | 0.0491 | 0.0489 | 0.0486 | 0.0491 |
| 17 | radon_redundant_chr | 0.0349 | 0.0343 | 0.0340 | 0.0349 | 0.0343 | 0.0340 | 0.0343 | 0.0343 | 0.0343 | 0.0343 | 0.0343 | 0.0343 |
| 18 | naive_bayes | 0.0941 | 0.0948 | 0.0955 | 0.0941 | 0.0948 | 0.0955 | 0.0953 | 0.0967 | 0.0962 | 0.0953 | 0.0967 | 0.0962 |
| 19 | mesquite_volume | 0.0206 | 0.0209 | 0.0208 | 0.0206 | 0.0209 | 0.0208 | 0.0194 | 0.0223 | 0.0198 | 0.0194 | 0.0223 | 0.0198 |
| 20 | cjs_t_t | 0.1015 | 0.0981 | 0.1015 | 0.1015 | 0.0981 | 0.1015 | 0.0960 | 0.0967 | 0.0978 | 0.0960 | 0.0967 | 0.0978 |
| 21 | irt_multilevel | 1.3379 | 0.9867 | 1.0794 | 1.3379 | 0.9867 | 1.0794 | 1.0716 | 0.9969 | 1.1124 | 1.0716 | 0.9969 | 1.1124 |
| 22 | irt | 1.1066 | 1.0423 | 0.9769 | 1.1066 | 1.0423 | 0.9769 | 1.1529 | 1.0222 | 1.0689 | 1.1529 | 1.0222 | 1.0689 |
| 23 | congress | 0.0277 | 0.0271 | 0.0272 | 0.0277 | 0.0271 | 0.0272 | 0.0273 | 0.0271 | 0.0280 | 0.0273 | 0.0271 | 0.0280 |
| 24 | dogs | 0.0454 | 0.0430 | 0.0447 | 0.0454 | 0.0430 | 0.0447 | 0.0436 | 0.0451 | 0.0447 | 0.0436 | 0.0451 | 0.0447 |
| 25 | Dynocc | 0.1224 | 0.1229 | 0.1329 | 0.1224 | 0.1229 | 0.1329 | 0.1212 | 0.1203 | 0.1209 | 0.1212 | 0.1203 | 0.1209 |
| 26 | multi_logit | 0.0981 | 0.0967 | 0.0964 | 0.0981 | 0.0967 | 0.0964 | 0.0982 | 0.0984 | 0.1002 | 0.0982 | 0.0984 | 0.1002 |
| 27 | electric_one_pred | 0.0209 | 0.0207 | 0.0213 | 0.0209 | 0.0207 | 0.0213 | 0.0206 | 0.0210 | 0.0212 | 0.0206 | 0.0210 | 0.0212 |
| 28 | election88 | 0.2777 | 0.2732 | 0.2747 | 0.2777 | 0.2732 | 0.2747 | 0.2398 | 0.2817 | 0.2668 | 0.2398 | 0.2817 | 0.2668 |
| 29 | wells_dist | 0.0753 | 0.0757 | 0.0775 | 0.0753 | 0.0757 | 0.0775 | 0.0765 | 0.0754 | 0.0759 | 0.0765 | 0.0754 | 0.0759 |
| 30 | wells | 0.0828 | 0.0819 | 0.0813 | 0.0828 | 0.0819 | 0.0813 | 0.0823 | 0.0816 | 0.0812 | 0.0823 | 0.0816 | 0.0812 |

Table 13: This table provides per iteration training times when using STL gradient from Equation (3) with our comprehensive step-search scheme. Please refer to Table 5 for lower-bound results.

| | $q_\phi$ family | Full-rank Gaussian | | | | | | | | | | | |
| | Step-search scheme | Comprehensive step-search | | | | | | | | | | | |
| | $\nabla_\phi$ | Estimate with STL | | | | | | | | | | | |
| | LI | Used | | | | | | Not Used | | | | | |
| | IWVI $M_{training}$ | 1 | | | | | | 1 | | | | | |
| | IWVI $M_{sampling}$ | 1 | | | 10 | | | 1 | | | 10 | | |
| | Independent Trial | Trial 1 | Trial 2 | Trial 3 | Trial 1 | Trial 2 | Trial 3 | Trial 1 | Trial 2 | Trial 3 | Trial 1 | Trial 2 | Trial 3 |
| Id | Model Name | | | | | | | | | | | | |
|---|---|---|---|---|---|---|---|---|---|---|---|---|---|
| 1 | lsat | 0.6813 | 0.6598 | 0.8223 | 0.6813 | 0.6598 | 0.8223 | 0.6002 | 0.9192 | 0.6543 | 0.6002 | 0.9192 | 0.6543 |
| 2 | Mh | 0.1398 | 0.1327 | 0.1330 | 0.1398 | 0.1327 | 0.1330 | 0.1295 | 0.1345 | 0.1340 | 0.1295 | 0.1345 | 0.1340 |
| 3 | test_simplex | 0.0191 | 0.0141 | 0.0129 | 0.0191 | 0.0141 | 0.0129 | 0.0213 | 0.0136 | 0.0259 | 0.0213 | 0.0136 | 0.0259 |
| 4 | endo3 | 0.0443 | 0.0315 | 0.0322 | 0.0443 | 0.0315 | 0.0322 | 0.0446 | 0.0310 | 0.0390 | 0.0446 | 0.0310 | 0.0390 |
| 5 | gp_predict | 1.1767 | 0.9317 | 0.8194 | 1.1767 | 0.9317 | 0.8194 | 1.1775 | 0.8208 | 1.1216 | 1.1775 | 0.8208 | 1.1216 |
| 6 | Mth | 0.1394 | 0.1194 | 0.1185 | 0.1394 | 0.1194 | 0.1185 | 0.1410 | 0.1418 | 0.1235 | 0.1410 | 0.1418 | 0.1235 |
| 7 | oxford | 0.0618 | 0.0486 | 0.0518 | 0.0618 | 0.0486 | 0.0518 | 0.0651 | 0.0527 | 0.0511 | 0.0651 | 0.0527 | 0.0511 |
| 8 | cjs_mnl | 0.0273 | 0.0210 | 0.0211 | 0.0273 | 0.0210 | 0.0211 | 0.0288 | 0.0210 | 0.0198 | 0.0288 | 0.0210 | 0.0198 |
| 9 | hepatitis | 0.0515 | 0.0500 | 0.0394 | 0.0515 | 0.0500 | 0.0394 | 0.0504 | 0.0416 | 0.0418 | 0.0504 | 0.0416 | 0.0418 |
| 10 | normal_multi | 0.1713 | 0.1477 | 0.1375 | 0.1713 | 0.1477 | 0.1375 | 0.1746 | 0.1475 | 0.1475 | 0.1746 | 0.1475 | 0.1475 |
| 11 | hiv_chr | 0.0319 | 0.0350 | 0.0352 | 0.0319 | 0.0350 | 0.0352 | 0.0421 | 0.0352 | 0.0348 | 0.0421 | 0.0352 | 0.0348 |
| 12 | electric_1c_chr | 0.0355 | 0.0283 | 0.0284 | 0.0355 | 0.0283 | 0.0284 | 0.0356 | 0.0283 | 0.0297 | 0.0356 | 0.0283 | 0.0297 |
| 13 | electric_1a_chr | 0.0323 | 0.0289 | 0.0271 | 0.0323 | 0.0289 | 0.0271 | 0.0330 | 0.0273 | 0.0289 | 0.0330 | 0.0273 | 0.0289 |
| 14 | electric_chr | 0.0299 | 0.0240 | 0.0244 | 0.0299 | 0.0240 | 0.0244 | 0.0304 | 0.0258 | 0.0245 | 0.0304 | 0.0258 | 0.0245 |
| 15 | radon_vary_si_chr | 0.0463 | 0.0377 | 0.0376 | 0.0463 | 0.0377 | 0.0376 | 0.0455 | 0.0373 | 0.0373 | 0.0455 | 0.0373 | 0.0373 |
| 16 | lda | 0.0478 | 0.0503 | 0.0495 | 0.0478 | 0.0503 | 0.0495 | 0.0479 | 0.0405 | 0.0406 | 0.0479 | 0.0405 | 0.0406 |
| 17 | radon_redundant_chr | 0.0308 | 0.0244 | 0.0241 | 0.0308 | 0.0244 | 0.0241 | 0.0308 | 0.0246 | 0.0245 | 0.0308 | 0.0246 | 0.0245 |
| 18 | naive_bayes | 0.0960 | 0.0855 | 0.0863 | 0.0960 | 0.0855 | 0.0863 | 0.0953 | 0.0762 | 0.0716 | 0.0953 | 0.0762 | 0.0716 |
| 19 | mesquite_volume | 0.0196 | 0.0145 | 0.0144 | 0.0196 | 0.0145 | 0.0144 | 0.0207 | 0.0145 | 0.0207 | 0.0207 | 0.0145 | 0.0207 |
| 20 | cjs_t_t | 0.0958 | 0.0821 | 0.0849 | 0.0958 | 0.0821 | 0.0849 | 0.0926 | 0.0845 | 0.0779 | 0.0926 | 0.0845 | 0.0779 |
| 21 | irt_multilevel | 0.8995 | 1.1413 | 1.1138 | 0.8995 | 1.1413 | 1.1138 | 0.8913 | 0.9329 | 0.7391 | 0.8913 | 0.9329 | 0.7391 |
| 22 | irt | 0.9916 | 0.6671 | 0.7762 | 0.9916 | 0.6671 | 0.7762 | 0.9206 | 0.7895 | 0.7731 | 0.9206 | 0.7895 | 0.7731 |
| 23 | congress | 0.0269 | 0.0183 | 0.0179 | 0.0269 | 0.0183 | 0.0179 | 0.0264 | 0.0179 | 0.0180 | 0.0264 | 0.0179 | 0.0180 |
| 24 | dogs | 0.0439 | 0.0369 | 0.0367 | 0.0439 | 0.0369 | 0.0367 | 0.0456 | 0.0367 | 0.0370 | 0.0456 | 0.0367 | 0.0370 |
| 25 | Dynocc | 0.1260 | 0.1156 | 0.1159 | 0.1260 | 0.1156 | 0.1159 | 0.1273 | 0.1155 | 0.1060 | 0.1273 | 0.1155 | 0.1060 |
| 26 | multi_logit | 0.0977 | 0.0822 | 0.0852 | 0.0977 | 0.0822 | 0.0852 | 0.0983 | 0.0842 | 0.0829 | 0.0983 | 0.0842 | 0.0829 |
| 27 | electric_one_pred | 0.0214 | 0.0146 | 0.0147 | 0.0214 | 0.0146 | 0.0147 | 0.0226 | 0.0155 | 0.0154 | 0.0226 | 0.0155 | 0.0154 |
| 28 | election88 | 0.2324 | 0.2754 | 0.2799 | 0.2324 | 0.2754 | 0.2799 | 0.2127 | 0.2765 | 0.2323 | 0.2127 | 0.2765 | 0.2323 |
| 29 | wells_dist | 0.0758 | 0.0587 | 0.0586 | 0.0758 | 0.0587 | 0.0586 | 0.0757 | 0.0606 | 0.0613 | 0.0757 | 0.0606 | 0.0613 |
| 30 | wells | 0.0817 | 0.0697 | 0.0680 | 0.0817 | 0.0697 | 0.0680 | 0.0812 | 0.0645 | 0.0673 | 0.0812 | 0.0645 | 0.0673 |

Table 14: This table provides per iteration training times for importance-weighted training for Gaussian $q_\phi$ optimized with our comprehensive step-search scheme. Please refer to Table 6 for lower-bound results.

| | | Estimated without dropping the score-function term | | | | | | Estimated with DReG | | | | | |
| | | Used | | | Not Used | | | Used | | | Not Used | | |
| | $q_\phi$ family | Full-rank Gaussian | | | | | | | | | | | |
| | Step-search scheme | Comprehensive step-search | | | | | | | | | | | |
| | $\nabla_\phi$ | | | | | | | | | | | | |
| | LI | | | | | | | | | | | | |
| | IWVI $M_{training}$ | 10 | | | 10 | | | 10 | | | 10 | | |
| | IWVI $M_{sampling}$ | 10 | | | 10 | | | 10 | | | 10 | | |
| | Independent Trial | Trial 1 | Trial 2 | Trial 3 | Trial 1 | Trial 2 | Trial 3 | Trial 1 | Trial 2 | Trial 3 | Trial 1 | Trial 2 | Trial 3 |
| Id | Model Name | | | | | | | | | | | | |
|---|---|---|---|---|---|---|---|---|---|---|---|---|---|
| 1 | lsat | nan | nan | nan | nan | nan | nan | 0.6082 | 0.6649 | 0.6574 | 0.6794 | 0.9469 | 0.6935 |
| 2 | Mh | 0.2637 | 0.2737 | 0.2745 | 0.2625 | 0.2658 | 0.2589 | 0.1373 | 0.1320 | 0.1278 | 0.1394 | 0.1295 | 0.1291 |
| 3 | test_simplex | 0.0137 | 0.0130 | 0.0122 | 0.0134 | 0.0127 | 0.0134 | 0.0129 | 0.0095 | 0.0096 | 0.0128 | 0.0086 | 0.0092 |
| 4 | endo3 | 0.0544 | 0.0560 | 0.0574 | 0.0556 | 0.0561 | 0.0580 | 0.0372 | 0.0264 | 0.0264 | 0.0374 | 0.0262 | 0.0283 |
| 5 | gp_predict | nan | 1.0351 | 1.0520 | 1.0954 | 1.0511 | 1.0693 | 0.9513 | 0.8739 | 0.9259 | 0.9786 | 0.9526 | nan |
| 6 | Mth | 0.2582 | 0.2711 | 0.2728 | 0.2607 | 0.2811 | 0.2593 | 0.1007 | 0.1323 | 0.1152 | 0.1336 | 0.1679 | 0.1089 |
| 7 | oxford | 0.0933 | 0.0876 | 0.0928 | 0.1013 | 0.0977 | 0.0979 | 0.0543 | 0.0440 | 0.0428 | 0.0598 | 0.0485 | 0.0510 |
| 8 | cjs_mnl | 0.0199 | 0.0209 | 0.0208 | 0.0208 | 0.0207 | 0.0207 | 0.0196 | 0.0156 | 0.0153 | 0.0202 | 0.0157 | 0.0157 |
| 9 | hepatitis | 0.0759 | 0.0719 | 0.0722 | 0.0787 | 0.0760 | 0.0760 | 0.0458 | 0.0427 | 0.0354 | 0.0455 | 0.0375 | 0.0374 |
| 10 | normal_multi | 0.1689 | 0.1717 | 0.1734 | 0.1699 | 0.1686 | 0.1707 | 0.1636 | 0.1412 | 0.1310 | 0.1628 | 0.1402 | 0.1419 |
| 11 | hiv_chr | 0.0557 | 0.0554 | 0.0546 | 0.0561 | 0.0557 | 0.0554 | 0.0275 | 0.0301 | 0.0302 | 0.0362 | 0.0303 | 0.0300 |
| 12 | electric_1c_chr | 0.0358 | 0.0358 | 0.0363 | 0.0361 | 0.0369 | 0.0366 | 0.0289 | 0.0230 | 0.0230 | 0.0289 | 0.0240 | 0.0240 |
| 13 | electric_1a_chr | 0.0331 | 0.0331 | 0.0332 | 0.0333 | 0.0332 | 0.0335 | 0.0258 | 0.0230 | 0.0204 | 0.0273 | 0.0222 | 0.0223 |
| 14 | electric_chr | 0.0285 | 0.0279 | 0.0300 | 0.0288 | 0.0297 | 0.0299 | 0.0228 | 0.0187 | 0.0193 | 0.0228 | 0.0194 | 0.0191 |
| 15 | radon_vary_si_chr | 0.0564 | 0.0568 | 0.0570 | 0.0585 | 0.0595 | 0.0575 | 0.0394 | 0.0347 | 0.0325 | 0.0392 | 0.0321 | 0.0321 |
| 16 | lda | 0.0411 | 0.0427 | 0.0410 | 0.0416 | 0.0419 | 0.0418 | 0.0411 | 0.0428 | 0.0417 | 0.0413 | 0.0333 | 0.0349 |
| 17 | radon_redundant_chr | 0.0276 | 0.0275 | 0.0273 | 0.0284 | 0.0280 | 0.0277 | 0.0237 | 0.0190 | 0.0188 | 0.0239 | 0.0188 | 0.0191 |
| 18 | naive_bayes | 0.0929 | 0.0876 | 0.0881 | 0.0878 | 0.0901 | 0.0889 | 0.0878 | 0.0795 | 0.0793 | 0.0883 | 0.0704 | 0.0667 |
| 19 | mesquite_volume | 0.0127 | 0.0128 | 0.0128 | 0.0132 | 0.0133 | 0.0133 | 0.0128 | 0.0097 | 0.0098 | 0.0128 | 0.0099 | 0.0096 |
| 20 | cjs_t_t | 0.0948 | 0.0913 | 0.0913 | 0.0949 | 0.0899 | 0.0889 | 0.0875 | 0.0764 | 0.0767 | 0.0889 | 0.0798 | 0.0725 |
| 21 | irt_multilevel | 1.0576 | 0.8762 | 0.9683 | 1.3115 | 0.9599 | 1.0815 | 0.9164 | 0.7620 | 0.8797 | 0.9028 | 0.7732 | 0.8005 |
| 22 | irt | 0.9248 | 1.0201 | 0.9852 | 1.0918 | 0.8881 | 0.9860 | 0.9761 | 0.5699 | 0.7827 | 0.9187 | 0.7041 | 0.7859 |
| 23 | congress | 0.0188 | 0.0190 | 0.0190 | 0.0199 | 0.0194 | 0.0201 | 0.0194 | 0.0130 | 0.0130 | 0.0191 | 0.0129 | 0.0130 |
| 24 | dogs | 0.0386 | 0.0362 | 0.0360 | 0.0373 | 0.0373 | 0.0372 | 0.0372 | 0.0295 | 0.0313 | 0.0368 | 0.0311 | 0.0311 |
| 25 | Dynocc | 0.1145 | 0.1155 | 0.1266 | 0.1150 | 0.1156 | 0.1137 | 0.1145 | 0.1104 | 0.1113 | 0.1181 | 0.1094 | 0.1001 |
| 26 | multi_logit | 0.0910 | 0.0901 | 0.0915 | 0.0920 | 0.0910 | 0.0911 | 0.0917 | 0.0796 | 0.0769 | 0.0916 | 0.0825 | 0.0782 |
| 27 | electric_one_pred | 0.0141 | 0.0149 | 0.0144 | 0.0138 | 0.0146 | 0.0149 | 0.0143 | 0.0100 | 0.0100 | 0.0140 | 0.0109 | 0.0108 |
| 28 | election88 | 0.2622 | 0.2345 | 0.2664 | 0.2677 | 0.2740 | 0.2724 | 0.2268 | 0.2689 | 0.2686 | 0.2063 | 0.2702 | 0.2688 |
| 29 | wells_dist | 0.0675 | 0.0690 | 0.0699 | 0.0689 | 0.0705 | 0.0688 | 0.0692 | 0.0528 | 0.0528 | 0.0681 | 0.0554 | 0.0549 |
| 30 | wells | 0.0749 | 0.0744 | 0.0741 | 0.0753 | 0.0743 | 0.0767 | 0.0750 | 0.0632 | 0.0624 | 0.0747 | 0.0590 | 0.0613 |

Table 15: This table provides per iterations training times for real-NVP normalizing flows optimized with our comprehensive step-search scheme with additional results from using IW-sampling. Please refer to Table 7 for lower-bound results.

| | $q_\phi$ family | Real NVP flows | | | | | | | | | | | |
|---|---|---|---|---|---|---|---|---|---|---|---|---|---|
| | Step-search scheme | Comprehensive step-search | | | | | | | | | | | |
| | $\nabla_\phi$ | Estimated without dropping the score-function term –full gradient | | | | | | Estimated with STL | | | | | |
| | LI | Not Used | | | | | | Not Used | | | | | |
| | IWVI $M_{training}$ | 1 | | | | | | 1 | | | | | |
| | IWVI $M_{sampling}$ | 1 | | | 10 | | | 1 | | | 10 | | |
| | Independent Trial | Trial 1 | Trial 2 | Trial 3 | Trial 1 | Trial 2 | Trial 3 | Trial 1 | Trial 2 | Trial 3 | Trial 1 | Trial 2 | Trial 3 |
| Id | Model Name | | | | | | | | | | | | |
| 1 | lsat | 1.1155 | 1.4013 | 0.8866 | 1.1155 | 1.4013 | 0.8866 | 0.8995 | 1.2909 | 1.3146 | 0.8995 | 1.2909 | 1.3146 |
| 2 | Mh | 0.4176 | 0.3657 | 0.3984 | 0.4176 | 0.3657 | 0.3984 | 0.3917 | 0.5270 | 0.5304 | 0.3917 | 0.5270 | 0.5304 |
| 3 | test_simplex | nan | nan | nan | nan | nan | nan | nan | nan | nan | nan | nan | nan |
| 4 | endo3 | 0.2509 | 0.2287 | 0.2311 | 0.2509 | 0.2287 | 0.2311 | 0.2562 | 0.2643 | 0.2549 | 0.2562 | 0.2643 | 0.2549 |
| 5 | gp_predict | 1.3029 | 1.0646 | 1.1239 | 1.3029 | 1.0646 | 1.1239 | 1.1514 | 1.0246 | 1.3611 | 1.1514 | 1.0246 | 1.3611 |
| 6 | Mth | 0.4158 | 0.3843 | 0.3718 | 0.4158 | 0.3843 | 0.3718 | 0.4255 | 0.5200 | 0.5311 | 0.4255 | 0.5200 | 0.5311 |
| 7 | oxford | 0.2828 | 0.2770 | 0.2678 | 0.2828 | 0.2770 | 0.2678 | 0.3053 | 0.2851 | 0.2996 | 0.3053 | 0.2851 | 0.2996 |
| 8 | cjs_mnl | 0.1424 | 0.1426 | 0.1459 | 0.1424 | 0.1426 | 0.1459 | 0.1936 | 0.1972 | 0.2079 | 0.1936 | 0.1972 | 0.2079 |
| 9 | hepatitis | 0.2471 | 0.2461 | 0.2467 | 0.2471 | 0.2461 | 0.2467 | 0.2720 | 0.2732 | 0.2724 | 0.2720 | 0.2732 | 0.2724 |
| 10 | normal_multi | 0.3088 | 0.3743 | 0.3030 | 0.3088 | 0.3743 | 0.3030 | 0.3247 | 0.3106 | 0.3868 | 0.3247 | 0.3106 | 0.3868 |
| 11 | hiv_chr | 0.2298 | 0.2091 | 0.2803 | 0.2298 | 0.2091 | 0.2803 | 0.2528 | 0.3103 | 0.3063 | 0.2528 | 0.3103 | 0.3063 |
| 12 | electric_1c_chr | 0.1899 | 0.2002 | 0.2004 | 0.1899 | 0.2002 | 0.2004 | 0.2110 | 0.1990 | 0.1888 | 0.2110 | 0.1990 | 0.1888 |
| 13 | electric_1a_chr | 0.1808 | 0.1958 | 0.1964 | 0.1808 | 0.1958 | 0.1964 | 0.2173 | 0.2198 | 0.2175 | 0.2173 | 0.2198 | 0.2175 |
| 14 | electric_chr | 0.1869 | 0.1794 | 0.1789 | 0.1869 | 0.1794 | 0.1789 | 0.2048 | 0.2053 | 0.2054 | 0.2048 | 0.2053 | 0.2054 |
| 15 | radon_vary_si_chr | 0.2396 | 0.2797 | 0.1970 | 0.2396 | 0.2797 | 0.1970 | 0.2552 | 0.2473 | 0.2442 | 0.2552 | 0.2473 | 0.2442 |
| 16 | lda | nan | 0.1483 | nan | nan | 0.1483 | nan | nan | nan | nan | nan | nan | nan |
| 17 | radon_redundant_chr | 0.1739 | 0.1640 | 0.1721 | 0.1739 | 0.1640 | 0.1721 | 0.1898 | 0.1959 | 0.1988 | 0.1898 | 0.1959 | 0.1988 |
| 18 | naive_bayes | 0.2366 | 0.2179 | 0.2147 | 0.2366 | 0.2179 | 0.2147 | 0.2365 | 0.2295 | 0.2384 | 0.2365 | 0.2295 | 0.2384 |
| 19 | mesquite_volume | 0.1289 | 0.1246 | 0.1249 | 0.1289 | 0.1246 | 0.1249 | 0.1508 | 0.1265 | 0.1484 | 0.1508 | 0.1265 | 0.1484 |
| 20 | cjs_t_t | 0.2017 | 0.2044 | 0.2041 | 0.2017 | 0.2044 | 0.2041 | 0.2159 | 0.2679 | 0.2090 | 0.2159 | 0.2679 | 0.2090 |
| 21 | irt_multilevel | 1.0242 | 1.1134 | 1.2535 | 1.0242 | 1.1134 | 1.2535 | 0.9318 | 1.1821 | 1.2016 | 0.9318 | 1.1821 | 1.2016 |
| 22 | irt | 1.0350 | 1.3052 | 1.1808 | 1.0350 | 1.3052 | 1.1808 | 1.1201 | 0.9608 | 1.1653 | 1.1201 | 0.9608 | 1.1653 |
| 23 | congress | 0.1310 | 0.1355 | 0.1354 | 0.1310 | 0.1355 | 0.1354 | 0.1832 | 0.1877 | 0.1922 | 0.1832 | 0.1877 | 0.1922 |
| 24 | dogs | 0.1495 | 0.1989 | 0.1571 | 0.1495 | 0.1989 | 0.1571 | 0.1617 | 0.1622 | 0.1631 | 0.1617 | 0.1622 | 0.1631 |
| 25 | Dynocc | 0.2708 | 0.2380 | 0.2273 | 0.2708 | 0.2380 | 0.2273 | 0.2886 | 0.2308 | 0.2843 | 0.2886 | 0.2308 | 0.2843 |
| 26 | multi_logit | 0.2261 | 0.2043 | 0.2153 | 0.2261 | 0.2043 | 0.2153 | 0.2235 | 0.2278 | 0.2367 | 0.2235 | 0.2278 | 0.2367 |
| 27 | electric_one_pred | 0.1318 | 0.1232 | 0.1317 | 0.1318 | 0.1232 | 0.1317 | 0.1776 | 0.1792 | 0.2418 | 0.1776 | 0.1792 | 0.2418 |
| 28 | election88 | 0.3636 | 0.3676 | 0.3965 | 0.3636 | 0.3676 | 0.3965 | 0.3134 | 0.3739 | 0.4694 | 0.3134 | 0.3739 | 0.4694 |
| 29 | wells_dist | 0.1663 | 0.2459 | 0.1756 | 0.1663 | 0.2459 | 0.1756 | 0.2180 | 0.2328 | 0.2324 | 0.2180 | 0.2328 | 0.2324 |
| 30 | wells | 0.1934 | 0.1830 | 0.1756 | 0.1934 | 0.1830 | 0.1756 | 0.2571 | 0.2318 | 0.1722 | 0.2571 | 0.2318 | 0.1722 |

Table 16: This table provides per iteration training times for importance weighted training for real-NVP normalizing flows optimized with our comprehensive step-search scheme with additional results from using regular IW-ELBO gradient. Please refer to Table 8 for lower-bound results.

| | | Real NVP flows | | | | | |
|---|---|---|---|---|---|---|---|
| | $q_\phi$ family | Real NVP flows | | | | | |
| | Step-search scheme | Comprehensive step-search | | | | | |
| | $\nabla_\phi$ | Estimated w/o dropping score-function term | | | Estimated with DReG | | |
| | LI | Not Used | | | Not Used | | |
| | IWVI $M_{training}$ | 10 | | | 10 | | |
| | IWVI $M_{sampling}$ | 10 | | | 10 | | |
| | Independent Trial | Trial 1 | Trial 2 | Trial 3 | Trial 1 | Trial 2 | Trial 3 |
| Id | Model Name | | | | | | |
| 1 | lsat | 1.3123 | 1.3801 | 0.8872 | 1.3382 | 1.1976 | 1.2966 |
| 2 | Mh | 0.4034 | 0.3471 | 0.3457 | 0.4056 | 0.5139 | 0.4923 |
| 3 | test_simplex | nan | nan | nan | nan | nan | nan |
| 4 | endo3 | 0.2403 | 0.2181 | 0.2214 | 0.2415 | 0.2480 | 0.2020 |
| 5 | gp_predict | 1.2093 | 1.3739 | 1.1092 | 1.0965 | 1.1590 | 1.2996 |
| 6 | Mth | 0.4022 | 0.3624 | 0.3600 | 0.3844 | 0.5129 | 0.4950 |
| 7 | oxford | 0.2711 | 0.2679 | 0.2685 | 0.2874 | 0.2765 | 0.2786 |
| 8 | cjs_mnl | 0.1344 | 0.1333 | 0.1365 | 0.1682 | 0.1452 | 0.1803 |
| 9 | hepatitis | 0.2365 | 0.2357 | 0.2366 | 0.2652 | 0.2565 | 0.2503 |
| 10 | normal_multi | 0.2988 | 0.3720 | 0.2505 | 0.3044 | 0.3897 | 0.2691 |
| 11 | hiv_chr | 0.2192 | 0.1919 | 0.2701 | 0.2368 | 0.2876 | 0.2792 |
| 12 | electric_1c_chr | 0.1797 | 0.1884 | 0.1895 | 0.1982 | 0.2019 | 0.1882 |
| 13 | electric_1a_chr | 0.1736 | 0.1854 | 0.1851 | 0.2022 | 0.1953 | 0.1974 |
| 14 | electric_chr | 0.1771 | 0.1756 | 0.1707 | 0.1911 | 0.2218 | 0.1918 |
| 15 | radon_vary_si_chr | 0.2386 | 0.2184 | 0.2069 | 0.2386 | 0.2347 | 0.2722 |
| 16 | lda | 0.1566 | 0.1510 | 0.1477 | 0.2008 | nan | 0.2252 |
| 17 | radon_redundant_chr | 0.1726 | 0.1641 | 0.1545 | 0.1913 | 0.1575 | 0.1830 |
| 18 | naive_bayes | 0.2059 | 0.2079 | 0.2053 | 0.2205 | 0.2174 | 0.2012 |
| 19 | mesquite_volume | 0.1174 | 0.1223 | 0.1155 | 0.1608 | 0.1679 | 0.1167 |
| 20 | cjs_t_t | 0.1875 | 0.1970 | 0.1942 | 0.2004 | 0.2206 | 0.1759 |
| 21 | irt_multilevel | 1.0148 | 1.1641 | 1.0829 | 1.0970 | 1.1833 | 1.1628 |
| 22 | irt | 1.1033 | 0.8217 | 1.1170 | 1.0006 | 0.9708 | 1.1640 |
| 23 | congress | 0.1214 | 0.1284 | 0.1267 | 0.1623 | 0.1694 | 0.1675 |
| 24 | dogs | 0.1407 | 0.1797 | 0.1925 | 0.1487 | 0.1559 | 0.1500 |
| 25 | Dynocc | 0.2327 | 0.2272 | 0.2273 | 0.2690 | 0.2357 | 0.2853 |
| 26 | multi_logit | 0.2138 | 0.1946 | 0.2068 | 0.2201 | 0.2298 | 0.2244 |
| 27 | electric_one_pred | 0.1209 | 0.1150 | 0.1201 | 0.1585 | 0.1612 | 0.1601 |
| 28 | election88 | 0.3934 | 0.3862 | 0.3583 | 0.3379 | 0.4366 | 0.3613 |
| 29 | wells_dist | 0.1652 | 0.2018 | 0.2176 | 0.1758 | 0.1630 | 0.2141 |
| 30 | wells | 0.2027 | 0.1720 | 0.1730 | 0.2212 | 0.2187 | 0.1587 |

Table 17: This table presents the per iteration training time for additional Diagonal Gaussian experiments. Please refer to Figure 11 and appendix B for more details.

| | $q_\phi$ family | Diagonal Gaussian | | | | | | | | |
|---|---|---|---|---|---|---|---|---|---|---|
| | Step-search scheme | Comprehensive step-search | | | | | | | | |
| | $\nabla_\phi$ | Closed form entropy | | | Estimated with STL | | | | | |
| | LI | Not Used | | | Not Used | | | | | |
| | IWVI $M_{training}$ | 1 | | | 1 | | | | | |
| | IWVI $M_{sampling}$ | 1 | | | 1 | | | 10 | | |
| | Independent Trial | Trial 1 | Trial 2 | Trial 3 | Trial 1 | Trial 2 | Trial 3 | Trial 1 | Trial 2 | Trial 3 |
| Id | Model Name | | | | | | | | | |
| 1 | lsat | 0.1602 | 0.1479 | 0.1616 | 0.7932 | 0.5420 | 0.5403 | 0.7932 | 0.5420 | 0.5403 |
| 2 | Mh | 0.0754 | 0.0682 | 0.0753 | 0.1278 | 0.1227 | 0.1240 | 0.1278 | 0.1227 | 0.1240 |
| 3 | test_simplex | 0.0138 | 0.0147 | 0.0150 | 0.0156 | 0.0147 | 0.0154 | 0.0156 | 0.0147 | 0.0154 |
| 4 | endo3 | 0.0274 | 0.0274 | 0.0288 | 0.0350 | 0.0347 | 0.0352 | 0.0350 | 0.0347 | 0.0352 |
| 5 | gp_predict | 0.8618 | 0.8190 | 1.0537 | 1.1967 | 1.2193 | 1.1846 | 1.1967 | 1.2193 | 1.1846 |
| 6 | Mth | 0.0719 | 0.0719 | 0.0702 | 0.1196 | 0.1231 | 0.1209 | 0.1196 | 0.1231 | 0.1209 |
| 7 | oxford | 0.0343 | 0.0339 | 0.0341 | 0.0471 | 0.0458 | 0.0485 | 0.0471 | 0.0458 | 0.0485 |
| 8 | cjs_mnl | 0.0178 | 0.0208 | 0.0193 | 0.0211 | 0.0207 | 0.0211 | 0.0211 | 0.0207 | 0.0211 |
| 9 | hepatitis | 0.0251 | 0.0279 | 0.0249 | 0.0388 | 0.0393 | 0.0386 | 0.0388 | 0.0393 | 0.0386 |
| 10 | normal_multi | 0.1237 | 0.1247 | 0.1244 | 0.1245 | 0.1481 | 0.1476 | 0.1245 | 0.1481 | 0.1476 |
| 11 | hiv_chr | 0.0261 | 0.0246 | 0.0261 | 0.0320 | 0.0332 | 0.0321 | 0.0320 | 0.0332 | 0.0321 |
| 12 | electric_1c_chr | 0.0250 | 0.0240 | 0.0245 | 0.0288 | 0.0274 | 0.0273 | 0.0288 | 0.0274 | 0.0273 |
| 13 | electric_1a_chr | 0.0230 | 0.0249 | 0.0216 | 0.0269 | 0.0264 | 0.0264 | 0.0269 | 0.0264 | 0.0264 |
| 14 | electric_chr | 0.0224 | 0.0210 | 0.0199 | 0.0242 | 0.0241 | 0.0241 | 0.0242 | 0.0241 | 0.0241 |
| 15 | radon_vary_si_chr | 0.0252 | 0.0285 | 0.0303 | 0.0352 | 0.0358 | 0.0351 | 0.0352 | 0.0358 | 0.0351 |
| 16 | lda | 0.0386 | 0.0374 | 0.0384 | 0.0395 | 0.0416 | 0.0414 | 0.0395 | 0.0416 | 0.0414 |
| 17 | radon_redundant_chr | 0.0200 | 0.0215 | 0.0201 | 0.0234 | 0.0239 | 0.0240 | 0.0234 | 0.0239 | 0.0240 |
| 18 | naive_bayes | 0.0736 | 0.0744 | 0.0658 | 0.0870 | 0.0859 | 0.0859 | 0.0870 | 0.0859 | 0.0859 |
| 19 | mesquite_volume | 0.0143 | 0.0140 | 0.0142 | 0.0145 | 0.0167 | 0.0145 | 0.0145 | 0.0167 | 0.0145 |
| 20 | cjs_t_t | 0.0860 | 0.0680 | 0.0831 | 0.0847 | 0.0756 | 0.0855 | 0.0847 | 0.0756 | 0.0855 |
| 21 | irt_multilevel | 0.6101 | 0.6449 | 0.6103 | 0.9602 | 0.9485 | 0.9637 | 0.9602 | 0.9485 | 0.9637 |
| 22 | irt | 0.6141 | 0.6213 | 0.5945 | 0.9598 | 0.7211 | 0.9360 | 0.9598 | 0.7211 | 0.9360 |
| 23 | congress | 0.0184 | 0.0181 | 0.0174 | 0.0185 | 0.0205 | 0.0185 | 0.0185 | 0.0205 | 0.0185 |
| 24 | dogs | 0.0347 | 0.0340 | 0.0358 | 0.0347 | 0.0337 | 0.0347 | 0.0347 | 0.0337 | 0.0347 |
| 25 | Dynocc | 0.1117 | 0.0915 | 0.0903 | 0.1054 | 0.1175 | 0.1164 | 0.1054 | 0.1175 | 0.1164 |
| 26 | multi_logit | 0.0776 | 0.0807 | 0.0719 | 0.0800 | 0.0851 | 0.0853 | 0.0800 | 0.0851 | 0.0853 |
| 27 | electric_one_pred | 0.0144 | 0.0164 | 0.0144 | 0.0154 | 0.0152 | 0.0154 | 0.0154 | 0.0152 | 0.0154 |
| 28 | election88 | 0.2186 | 0.2183 | 0.2189 | 0.2808 | 0.2346 | 0.2838 | 0.2808 | 0.2346 | 0.2838 |
| 29 | wells_dist | 0.0507 | 0.0569 | 0.0538 | 0.0587 | 0.0592 | 0.0587 | 0.0587 | 0.0592 | 0.0587 |
| 30 | wells | 0.0652 | 0.0602 | 0.0724 | 0.0671 | 0.0685 | 0.0667 | 0.0671 | 0.0685 | 0.0667 |