[Reviews · NeurIPS 2020]

Review 1

Summary and Contributions: The authors study the empirical advantage from recent improvements in black-box VI on probabilistic models from the Stan model library. They specifically consider the effect of step-size search, the sticking the landing (STL) gradient estimator, using importance weighting in the training objective and/or at test time, and the use of RealNVP flows as a more flexible approximating posterior. Combining these significantly outperforms naive ADVI; most techniques appear to be individually helpful, except for importance-weighted training, which gives inconsistent results.

Strengths: Stipulating that the idea of using (and combining) these techniques was likely 'obvious' to anyone working in the field---there is no novel theoretical ground here---the empirical analysis in this paper is careful, convincing, and quite valuable. It considers the most prominent techniques recently proposed for black-box VI, and gives clear evidence of their value (or lack thereof) on a representative set of real-world inference tasks. This should provide developers of probabilistic programming systems with a clarified roadmap, as well as providing inference researchers with a modern set of baselines.

Weaknesses: Any empirical comparison is going to have the flaw of being insufficiently exhaustive, and this one is no exception. For example: - ADVI as implemented in Stan encompasses both full-covariance and diagonal Gaussian surrogates, but this paper evaluates only one of those, and it wasn't even clear which one until quite far in (line 297). This should be clarified earlier. Ideally it would be nice to see the relative performance of both Gaussian baselines (and perhaps other commonly-suggested schemes like a diagonal + low rank covariance). - I've seen IAF-style flows more commonly than RealNVP the past few years. Was RealNVP chosen because it supports sticking-the-landing? It would be useful to see a side-by-side comparison against a similar-size IAF without sticking-the-landing. - A simple method not included (maybe because it's so simple that no one has published on it for VI recently) is Polyak-Ruppert averaging, i.e., averaging the variational parameters over the final steps of stochastic optimization. One might hypothesize that this could yield some of the same benefits as the sticking the landing estimator (the average of noisy optimization steps is much less noisy than any individual step) while being more general and easier to implement; it'd be interesting to see if that's the case. A lot of care was put into evaluating importance-weighted objectives on a fair computational playing field. One might also expect there to be differences in time needed to train a Gaussian (diagonal or full-covariance) versus a normalizing flow, but I didn't see this discussed at all. How did the flow training times compare? Was there a time<->quality tradeoff in the capacity of the flow? Since a full-covariance Gaussian has O(n^2) parameters, at some problem size you'd expect a flow with constant-size layers to be faster than the Gaussian---was this encountered? It's not obvious that the ELBO makes sense as the sole objective to report; at least in principle a method could succeed in improving the ELBO while actually making predictions worse. Was there any analysis of how the ELBO correlates with other metrics of interest; for example, accuracy of estimated posterior moments, or predictive likelihood of held-out data?

Correctness: Subject to the points above, I thought the techniques chosen and the experiments conducted were reasonable.

Clarity: The paper is well written and easy to follow. nits: abstract line 9-10: "for which there are no clear guidance" should probably be "for which there is no clear guidance" line 333: "performs provides"

Relation to Prior Work: The entire paper is a discussion of previous contributions. (so, yes).

Reproducibility: Yes

Additional Feedback:


Review 2

Summary and Contributions: This work makes the necessary and meticulous exercise of exploring and optimizing for the best set of algorithmic components to use for black-box variational inference. The goal is for the practitioner not to have to tinker with inference details. --- UPDATE: Thank you for the rebuttal. I do not wish to change my evaluation. I recommend this paper for acceptance.

Strengths: The main strength of this work is its careful and thorough study of algorithmic components to improve off-the-shelf black-box VI. The final method combines importance sampling, normalizing flows, a robust step-size scheme and the STL gradient estimator to improve performance by one nat or more on a least 60% of the benchmark models when compared to ADVI. The study proceeds incrementally, adding components one by one, hence making it easy to assess the marginal benefits of each component in combination to the previous others. While this work does not contribute any new algorithmic component by itself, I find it important and critical that some in our community take a step back and evaluate how recent developments ought to be best combined in order to be the most helpful for the scientists using the tools we develop. This exercise is far from being trivial to carry out properly -- yet this study follows a strict methodology and successfully identifies several concrete recommendations that lead to significant improvements on average.

Weaknesses: While visualizing and reporting results across 30 benchmarks and multiple variants of a set of algorithmic components is challenging, I would have appreciated if the analysis had come with an assessment of the variability of the performance improvements.

Correctness: Yes. The methodology appears to be correct, although the supplementary materials are full of additional details which I have not checked.

Clarity: Yes, the paper is clearly written and easy to follow for someone familiar with the components that are evaluated.

Relation to Prior Work: Yes.

Reproducibility: Yes

Additional Feedback:


Review 3

Summary and Contributions: The paper studies the best practice of using ADVI with some recent advances in related fields. Specifically, the paper examines 1) the way to search step size, 2) the choice of gradient estimators, 3) variational parameter initialisations, 4) different lower bounds of log-likelihood and 5) replacing Gaussian by normalising flows (NFs) as the variational posterior. The paper performs well-design experiments on 30 Stan models to study each option and the effect of combing them. The paper then concludes the best-practice: using NFs via a regular ELBO under the STL gradient estimator (training) and using importance sampling during inference.

Strengths: The goal of the paper is simple but useful. It answers a few related questions: what's the best practice of using ADVI with the recent new techniques related, how much gain can we get by applying these techniques (together) and how reliable are they? Those are real questions users would meet and would be genuinely nice to have a guide to pave the way of wider use of all these techniques as well as ADVI. How the paper tackles the reliability part by considering a good collection of models is good and the and visualisations are informative. Although there is no novelty of the methods themselves, I think the paper would be useful for a wide range of practitioners.

Weaknesses: No novelty of the methodology. The paper misses a few related works [1,2] and misses a conclusion/discussion section. [1] Structured Conditional Continuous Normalizing Flows for Efficient Amortized Inference in Graphical Models, AISTATS 2020 (http://proceedings.mlr.press/v118/fjelde20a.html) [2] Bijectors.jl: Flexible transformations for probability distributions, AABI 2019 (http://proceedings.mlr.press/v118/fjelde20a.html) [1] is on how to take use of the structure of NFs for a better variational approximation. [2] has a similar but much simple example on the same idea in the context of ADVI.

Correctness: The methodology of conducting the experiments looks correctly to me and the final conclusion makes sense.

Clarity: The paper is very well-written and easy to follow. Although as I mentioned, there seems no conclusion/discussion section.

Relation to Prior Work: I mentioned a few papers the author(s) fail(s) to link on the topic of applying NFs to ADVI, or automated VI (or amortised inference). There might be some more in the field as I'm not closely following it. Despite of this fact, the focus of the paper is quite difference from those I pointed out.

Reproducibility: Yes

Additional Feedback: The Broader Impact section is too vague. One potential concern of so-called "best practice" is that we would also like to know when they might fail so that we can avoid applying them. As from the results, there are still a small fraction of models which has a degraded performance after applying all the techniques - can we take a careful look on them - are they in a special model family? Ideally, questions like this should be addressed in the main paper, as claiming something "the default" would affect a lot of real users. ######### # Update # ######### The author feedback looks good to me and I keep my original score.

[Author Response · NeurIPS 2020]

We are grateful for the reviewers' detailed comments. We will rectify all small writing errors, and offer comments for
more significant issues below.

@ **R1 – Other Gaussian baselines:** We concentrated on full-rank Gaussians because these give dramatically better
results than diagonal Gaussians; we will make sure to clarify this earlier on. With minimal space, we did not want
emphasize the issue of full-rank vs. diagonal Gaussians, since this is a (relatively) well covered issue. In retrospect,
however, we definitely appreciate your point that this would be valuable to see in the context of our full experiments. We
will at a minimum perform a comparison to diagonal Gaussians and add the results to the appendix. (We are concerned
about a lack of space for a full discussion in the main paper.) The summary is that diagonal Gaussians perform much
worse!

@ **R1 – Time tradeoffs :** Thank you for pointing out that this discussion needs attention. Both of the your suspicions
are correct. First, in preliminary experiments, we found an increase in the number of coupling layers correlated with
better performance. So there is indeed a time-quality tradeoff there. (We do not attempt to find the best tradeoff as
our goal is not really to find the best possible flow.) This should be clarified. Second, it is true that the Gaussian
scales as $O(n^2)$ while flows scale as $O(n)$. After some calculation, one can show that the number of parameters for
Gaussian will exceed that for RealNVP if the latent dimensions exceed 2000; (we discuss this briefly in Section F,
the last paragraph.) While flows scale better, the computational constant is also higher. At the same time, when the
dataset is large, the cost of $\log p(z,x)$ dominates costs involving the variational distribution. And further, our code is
implemented in Python while models are implemented in Stan, which complicates the meaning of running times in low
dimensions when interpretation overhead is a factor. Still, we accept that running times measurements are helpful even
if provisional, and will add these to the appendix.

@ **R1 – ELBO correlation with other metrics:** Of course, an ELBO improvement is equivalent to an improvement
in KL divergence between a variational approximation and the true posterior. Several works have reported an empirical
correlation between the ELBO improvements with improvements in test-likelihoods (see [21, Appendix, Table 3, 4, and
5]; Tao et al., 2018, Figure 4; Mishkin et al., 2018, Figure 4.) and accuracy of posterior moments (see [9, Figure 6]; [10,
Figure 7, and Figure 9 to 22]).) We expect our methods to follow similar correlations for ELBO improvements. The
advantage of using an ELBO is that the results are far more stable since they do not depend on details like train-test
splits, or the particular metric of accuracy.

@ **R1 – IAF-style flows and Polyak-Ruppert averaging:** Yes, as suspected, we use RealNVP because it is reason-
ably "generic" and supports the STL estimator; we will make this clear in the paper. We agree that a more exhaustive
comparison of possible flows would indeed be useful, but may not be the highest priority given space limitations. We are
intrigued by the hypothesis about Polyak-Ruppert averaging, but must be honest that it may be challenging to include.

@ **R2 – Assessment of Variability:** Thank you for mentioning the challenge of statistical measure of variability in
empirical CDFs. We considered this issue at length while writing the paper. There are standard methods for calculating
confidence intervals, but applied naively, and these would not do what we want. (They would estimate generalization to
other "models" (drawn from the same distribution of models) rather than generalizing to running the same inference
methods with different random numbers.) Instead, we settled on the simplicity and transparency of running three
completely independent experiments for each change and superimposing the results. We believe this is adequate because
the variability is minimal in almost all cases. (The only exceptions are due to the ADVI step size scheme; however, the
improvements are so vast that the conclusions are not in doubt.) We do provide full results for all independent trials in
Appendix J and will add more discussion to the paper.

@ **R3 – Prior Work and writing suggestions:** Thank you for pointing out the two missed papers–we will discuss
them. We will also make the Broader Impact section more specific. Regarding the "default choice," it is certainly true
that a small number of models exist on which the changes degrade performance. We will better acknowledge this.
However, we suspect that it may be difficult to draw reliable conclusions about what models are at issue. Concretely,
compare ADVI against the best-performing approach (4c). The set of models where ADVI performs better is precisely
one: `gp-predict`. We certainly agree that our analysis is not the "last" word. We think that with more innovation in
analysis techniques (and a larger corpus) model-family specific details can be teased-out; still, this paper represents a
first step in the direction of rigorous empirical evaluation for inference research. Lastly, we will make the discussion in
Appendix C more prominent.

Tao et al. Variational Inference and Model Selection with Generalized Evidence Bounds. ICML, 2018

Mishkin et al. SLANG: Fast Structured Covariance Approximations for Bayesian Deep Learning with Natural Gradient. NeurIPS
2018.

[Meta-Review · NeurIPS 2020]

The authors study recent improvements of black-box VI and study the effect of step-size search, the sticking the landing (STL) gradient estimator, using importance weighting in the training objective and/or at test time, and the use of RealNVP flows as a more flexible approximating posterior. Combining these significantly outperforms naive ADVI. Strengths: - the empirical evaluation is careful and convincing - 30 benchmarks were considered Weaknesses: - no new method was proposed